# Is a more physical representation of aerosol activation needed for simulations of fog?

Craig Poku[1,a], Andrew N. Ross[1], Adrian A. Hill[2], Alan M. Blyth[1,3], and Ben Shipway[2]

[1]School of Earth and Environment, University of Leeds, Leeds, UK
[2]Met Office, Exeter, UK
[3]National Centre of Atmospheric Sciences, University of Leeds, Leeds, UK
[a]now at: Wolfson Atmospheric Chemistry Laboratories, University of York, York, UK

**Correspondence:** Craig Poku (craig.poku@york.ac.uk)

**Abstract.** Aerosols play a crucial role in the fog life cycle, as they determine the droplet number concentration, and hence droplet size, which in turn controls both the fog's optical thickness and life span. Detailed aerosol-microphysics schemes which accurately represent droplet formation and growth are unsuitable for weather forecasting and climate models, as the computational power required to calculate droplet formation would dominate the treatment of the rest of the physics in the
model. A simple method to account for droplet formation is the use of an aerosol activation scheme, which parameterises the droplet number concentration based on a change in supersaturation at a given time. Traditionally, aerosol activation parameterisation schemes were designed for convective clouds and assume that supersaturation is reached through adiabatic lifting, with many imposing a minimum vertical velocity (e.g. $0.1$ m s$^{-1}$) to account for the unresolved sub-grid ascent. In radiation fog, the measured updrafts during initial formation are often insignificant, with radiative cooling being the dominant process
leading to saturation. As a result, there is a risk that many aerosol activation schemes will overpredict the initial fog droplet number concentration, which in turn may result in the fog transitioning to an optically thick layer too rapidly.

    This paper presents a more physically-based aerosol activation scheme that can account for a change in saturation due to non-adiabatic processes. Using an offline model, our results show that the equivalent cooling rate associated with the minimum updraft velocity threshold assumption can overpredict the droplet number by up to 70% in comparison to a typical cooling
rate found in fog formation. The new scheme has been implemented in the Met Office Natural Environment Research Council (NERC) Cloud (MONC) LES model and tested using observations of a radiation fog case study based in Cardington, UK. The results in this work show that using a more physically-based method of aerosol activation leads to the calculation of a more appropriate cloud droplet number. As a result, there is a slower transition to an optically thick (well-mixed) fog that is more in line with observations.

The results shown in this paper demonstrate the importance of aerosol activation representation in fog modelling and the impact that the cloud droplet number has on processes linked to the formation and development of radiation fog. Unlike the previous parameterisation for aerosol activation, the revised scheme is suitable to simulate aerosol activation in both fog and convective cloud regimes.

# 1 Introduction

Fog can be defined as a cloud at ground level with a surface visibility of less than 1 km (WMO, 1966). It can cause major disruption to road, aviation and marine transport, with associated economic losses that are comparable to those resulting from winter storms and hurricanes (Gultepe et al., 2007). Fog can have negative impacts on human health and the safety of certain activities. For example, thick fog on 5th September 2013 resulted in the Sheppey crossing crash in southeast England, consequently injuring 60 people (BBC, 2013). Understanding the physics behind fog is crucial in improving fog forecasting

and mitigating the impact of such events.

An uncertainty within fog forecasting is caused by aerosol-fog interaction representation (Pruppacher and Klett, 2010). Aerosols are important for both clouds and fog, as they act as the substrate on which water condenses and droplets form. The growth rate of these droplets is dependent on the initial aerosol size and solubility. The aerosols are considered to be 'activated' once these droplets reach a certain size, where they can grow more easily within a saturated environment (known as cloud con-

densation nuclei (CCN)). The aerosol population is split by size categories. These size categories (hereafter known as modes) are technically defined as: the Aitken mode, where the diameter, $d$, of an aerosol particle is $< 0.1$ $\mu$m; the accumulation mode, where $0.1 \leq d \leq 1.0$ $\mu$m; and the coarse mode, where $d > 1.0$ $\mu$m (Whitby, 1978). Due to their size, Aitken mode aerosols have an increased tendency to coagulate with other particles and not activate in their own right. In contrast, accumulation and coarse mode aerosols can activate into fog droplets, therefore indirectly impacting the cloud's microphysical structure and its

life span (e.g. Twomey, 1974; Albrecht, 1989). These impacts have been studied in great depth over the last few decades, both in the context of climate (e.g. IPCC, 2001) and meteorology (e.g. Seifert and Heus, 2013; Miltenberger et al., 2018). While research into radiation fog spans the last 100 years (e.g. Taylor, 1917; Roach et al., 1976), studies investigating aerosol impacts on fog are more recent. For example, Bott (1991) shows that aerosols fundamentally control radiation fog's optical thickness, and additional studies (e.g. Stolaki et al., 2015; Maalick et al., 2016) have verified why it's critical to correctly represent

different aerosol indirect effects when simulating fog.

Accurate droplet nucleation representation, i.e. aerosol activation, is essential to represent the aerosol indirect effects on clouds. However, when investigating aerosol-cloud interactions in models such as general circulation models (GCMs) and numerical weather prediction (NWP) models, many detailed droplet growth schemes are unsuitable, as the computational power required would dominate the treatment of the rest of the physics in the model (Ghan et al., 1993). Original development

of an aerosol activation parameterisation began by Squires (1958), with work by Twomey (1959) expanding on the modelling of aerosol activation. Twomey (1959) discussed the link between an aerosol spectrum, supersaturation and droplet number concentration. Using Köhler Theory, Twomey (1959) formulated a parameterisation based on the change in supersaturation for a given time, such that:

$$\frac{ds}{dt} = \alpha - \beta s \int\limits_0^s \nu(\sigma) \left[ \int\limits_{\tau(\sigma)}^t s dt \right]^{\frac{1}{2}} d\sigma, \tag{1}$$

where $\alpha$ is the supersaturation source due to atmospheric cooling, with the second term of Eq. (1) representing water vapour condensation onto the activated aerosol population. The constant, $\beta$, is dependent on the aerosol spectrum, with $\nu(\sigma)\delta\sigma$ being the number of nuclei in a unit volume with critical supersaturation between $\sigma$ and $\sigma + \delta\sigma$. As condensation results in a decrease in supersaturation, the maximum number of activated aerosols is capped and will occur once the peak supersaturation is reached (i.e. when the condensation term starts to dominate the cooling terms), resulting in no more aerosols activating. At this point,

$\frac{ds}{dt} = 0$, and Eq. (1) becomes:

$$\alpha = \beta s \int_0^s \nu(\sigma) \left[ \int_{\tau(\sigma)}^t s \, dt \right]^{\frac{1}{2}} d\sigma. \tag{2}$$

Different authors have addressed solving the right-hand side of Eq. (2). Twomey (1959) formulated an upper and lower bound to the inner integral in Eq. (2) and assumed an aerosol spectrum, which was later developed further by Cohard et al. (1998), Shipway and Abel (2010) and Shipway (2015). Ghan et al. (1993) developed a scheme that accounted for a more realistic

aerosol size distribution, which was naturally bounded by the total aerosol number. They showed that accounting for a more realistic single-mode aerosol-size distribution (lognormal) improved the parameterised number of droplets activated. However, because droplet growth was neglected upon activation in their scheme, the introduction of multi-mode aerosol resulted in big discrepancies between the explicit and parameterised number of activated droplets. Work by Abdul-Razzak et al. (1998) (and later Abdul-Razzak and Ghan, 2000) combined the benefits of the parameterisations developed by both Twomey (1959) and

Ghan et al. (1993). The scheme was not only bound by the total aerosol number but also assumed that growth continued from the point of activation. The result of these assumptions led to the parameterised number of activated aerosols agreeing better with the explicit calculation for activation, even in regimes of high updraft velocities (Abdul-Razzak and Ghan, 2000). There has also been work to move away from using aerosol activation schemes in fog simulations using large eddy simulations. Recent work by Schwenkel and Maronga (2019) has shown that the choice in condensation calculation can be critical when

investigating aerosol-fog interactions using large eddy simulations. More specifically, the same authors followed this study by demonstrating that using a bulk microphysics scheme in comparison to a Lagrangian cloud model (LCM) can overestimate liquid water and inaccurately represent the fog droplet distribution (Schwenkel and Maronga, 2020). However, using methods such as LCMs is unsuitable for weather and climate models due to their massive computational expense.

So far, the activation schemes discussed that are suitable for weather and climate models (i.e. Cohard et al., 1998; Abdul-

Razzak et al., 1998; Abdul-Razzak and Ghan, 2000; Shipway, 2015) have been tested assuming that saturation is driven by adiabatic ascent. In addition, a number of the listed schemes impose a fixed minimum updraft velocity threshold, $w_{min}$, of 0.1 m s$^{-1}$, corresponding to a cooling rate of 3.51 K hr$^{-1}$ assuming a dry adiabatic lapse rate (e.g. Ghan et al., 1997; Abdul-Razzak and Ghan, 2000; Morrison and Gettelman, 2008; West et al., 2014). A $w_{min}$ is suitable for these schemes, as they are designed to consider updrafts found in stratocumulus and convective clouds (Abdul-Razzak and Ghan, 2000; Meskhidze et al., 2005).

Furthermore, some models (such as GCMs) will use the subgrid velocity (derived from the subgrid turbulence) to calculate the number of droplets. However, the turbulence driven by cloud-top radiative cooling can be poorly resolved above the planetary

boundary layer (PBL) unless the model's vertical resolution was $< 100$ m (Ghan et al., 1997). Since such resolutions are not feasible in operational NWP or climate models, a $w_{min}$ of $0.1$ m s$^{-1}$ is imposed to account for this unresolved turbulence (Ghan et al., 1997). In radiation fog, the main mechanism for the initial formation of droplets is radiative cooling; a non-adiabatic process, with measured cooling rates of 1 - 4 K hr$^{-1}$ at the surface (calculated using data from Price, 2011) and updraft velocities close to 0 m s$^{-1}$. Consequently, both the assumption of saturation being driven by adiabatic ascent, and the use of a minimum vertical velocity threshold do not accurately account for aerosol activation in fog (as discussed in Boutle et al., 2018). Finally, although there are studies that focus on investigating using a non-adiabatic framework in aerosol activation schemes when simulating fog (e.g. Zhang et al., 2014; Schwenkel and Maronga, 2019), there are no studies to the authors' knowledge that test these assumptions for fog formation in clean aerosol regimes. Therefore, this may mean that using their schemes to simulate rural fog cases may lead to an overestimation in condensation (Shipway, 2015).

This paper will focus on addressing the assumptions using in activation scheme schemes to simulate fog with the modified Shipway (2015). It was chosen to use Shipway over Abdul-Razzak and Ghan (2000) (hereafter referred to as ARG), as it has been shown that ARG overestimates condensation in low aerosol regimes, making it activate too few aerosols (Shipway, 2015). The work presented in this paper has been split into two sections: firstly comparing the original Shipway scheme (henceforth Shipway) with the modified Shipway scheme developed here (SMOD) using an offline box model, and secondly comparing both of these schemes using large eddy simulations (LES) of an idealised fog case study (as described in Poku et al., 2019). During both comparisons, the following questions will be addressed:

1. What are the potential differences in aerosol activation between the Shipway and SMOD scheme?

2. How do the differences in aerosol activation representation impact the fog evolution in a large eddy simulation?

3. What potential discrepancies are not accounted for when simulating aerosol-fog interactions?

Section 2 will present how the Shipway and SMOD scheme differs from each other mathematically. Section 3 will outline the Shipway box model setup and how the SMOD was implemented into it. Section 4 addresses research question 1. Section 5 describes the LES model used and addresses research question 2. A discussion and conclusion will then follow.

## 2 SMOD - Modifying the Shipway activation scheme to include non-adiabatic cooling

### 2.1 Shipway activation scheme

The Shipway (2015) aerosol activation scheme is designed as an improvement to the original lower bound approximation by Twomey (1959), and utilises a lookup table method that solves the maximum supersaturation at a reduced computational expense. Shipway assumes the differential activity spectrum, $\phi(s)$, to be lognormal, which can be expressed as:

$$\phi(s) = \sum_{i=1}^{I} \frac{N_i}{\sqrt{2\pi}\ln(\sigma_{s,i})s}\exp\left(-\frac{\ln^2(s/s_{0,i})}{\ln^2\sigma_{s,i}}\right), \tag{3}$$

where $N_i$ is the number concentration of dry aerosol, $\sigma_{s,i}$ is the standard deviation of the distribution of $\phi(s)$, and $s_{0,i}$ is the mean geometric supersaturation for each given aerosol mode. Shipway (2015) formulated a new expression for the maximum supersaturation using the original Twomey (1959) lower bound approximation, such that:

$$\frac{\sqrt{2}\alpha^{\frac{3}{2}}}{\gamma} = s_{\max} \int_0^{s_{\max}} \phi(\sigma) \left[ \frac{1}{2}\left(1 - \left(\frac{\sigma}{s_{\max}}\right)^{\mu}\right)^{\lambda}\right]^{-1} \left(s_{\max}^2 - \sigma^2\right)^{\frac{1}{2}} d\sigma, \tag{4}$$

where $\mu$ and $\lambda$ are chosen empirically by Shipway (2015) such that $\mu = 3$ and $\lambda = 0.6$. $\alpha$ relates to the increase in relative humidity and hence saturation, due to an air parcel undergoing atmospheric cooling. To date, the Shipway activation scheme assumes that $\alpha$ is driven by an updraft velocity, i.e.

$$\alpha = \psi(T,p)\frac{dz}{dt}, \tag{5}$$

where $\psi(T)$ is the thermodynamical function associated with a change in supersaturation and pressure due to adiabatic
ascent, with:

$$\psi = \frac{c_p}{R_a T} - \frac{L}{R_v T^2}, \tag{6}$$

$L$ being the specific latent heat of vaporisation, and $\gamma$ being a temperature pressure variable related to the change in temperature due to latent heat release, such that:

$$\gamma = \frac{p}{\epsilon e_s} + \frac{L^2}{R_v c_p T^2}. \tag{7}$$

Using a precalculated lookup table to solve the right-hand side of Eq. (4) and again $s_{\max}$, Shipway (2015) calculates the total number of activated aerosols, $N_{act}$:

$$N_{act} = \frac{N_i}{2}\left[1 + \text{erf}\left(\frac{\ln(s_{\max}/s_{0,i})}{\sqrt{2}\ln\sigma_{s,i}}\right)\right], \tag{8}$$

with $\text{erf}(x)$ being the error function (Abramowitz and Stegun, 1965). For the SMOD scheme (see Appendix A for further details), the term, $\alpha$, in Eq. (4) has been modified to account for non-adiabatic cooling, such that:

$$\alpha = \psi_1 \left.\frac{dT}{dt}\right|_{ad} + \psi_2 \left.\frac{dT}{dt}\right|_{non\_ad}, \tag{9}$$

**Table 1.** Aerosol properties used to test the Shipway and SMOD schemes in the Shipway box model (Whitby, 1978).

| Environmental setting | Distribution parameters | Aitken mode | Accumulation mode | Coarse mode |
|---|---|---|---|---|
| Marine | N (cm $^{-3}$) | 340 | 60 | 3.1 |
| | $\sigma$ | 1.6 | 2.0 | 2.7 |
| | r ($\mu$m) | 0.005 | 0.035 | 0.31 |
| Clean continental | N (cm $^{-3}$) | 1000 | 800 | 0.72 |
| | $\sigma$ | 1.6 | 2.1 | 2.2 |
| | r ($\mu$m) | 0.008 | 0.034 | 0.46 |
| Urban | N (cm $^{-3}$) | 10600 | 32000 | 5.4 |
| | $\sigma$ | 1.8 | 2.16 | 2.21 |
| | r ($\mu$m) | 0.007 | 0.027 | 0.43 |

where:

$$\psi_1 = \frac{c_p}{R_a T} - \frac{L}{R_v T^2},$$

$$\psi_2 = -\frac{L}{R_v T^2}. \tag{10}$$

The SMOD scheme differs from Shipway when calculating $N_{\mathrm{act}}$, in that it uses Eq. (9) to solve $s_{\mathrm{max}}$ (see Table A1 in Shipway, 2015, for a summary of terms described in this section and Appendix A for a full derivation of $\psi_{1,2}$). This term has also been used in previous studies such as Schwenkel and Maronga (2019) when investigating nocturnal radiation fog using LES.

## 3 The Shipway box model - offline setup

To understand the flexibility of the SMOD scheme and how the thermodynamical function associated with the non-adiabatic contribution may impact $N_{\mathrm{act}}$, both the Shipway and extended SMOD activation schemes will be directly compared using the Shipway box model (Shipway, 2015). The Shipway box model is designed as a non-interactive offline suite to calculate the initial number of activated aerosols in a range of different environmental settings. As the model is non-interactive, it permits analysis of parameter space, in the absence of atmospheric feedbacks. Inputs of the model are potential temperature, vertical velocity and aerosol population properties (number concentration, size, mode and distribution size parameters). Shipway (2015) used the box model to test the Shipway (2015) and Twomey (1959) activation schemes in different aerosol regimes, in addition to schemes developed by Abdul-Razzak and Ghan (2000) and Nenes and Seinfeld (2003).

**Table 2.** The tests conducted in the offline box model that directly compared Shipway and SMOD adiabatic mode based on Eq.'s (4) and (9).

| Case | Tests in case | Scheme used | Aerosol mode | Environment |
|------|---------------|-------------|--------------|-------------|
| C_ship_ad_mar | T_ship_mar_ait | Shipway | Aitken | Marine |
|  | T_ship_mar_acc |  | Accumulation |  |
|  | T_ship_mar_coa |  | Coarse |  |
| C_ship_ad_con | T_ship_con_ait | Shipway | Aitken | Clean Continental |
|  | T_ship_con_acc |  | Accumulation |  |
|  | T_ship_con_coa |  | Coarse |  |
| C_ship_ad_urb | T_ship_urb_ait | Shipway | Aitken | Urban |
|  | T_ship_urb_acc |  | Accumulation |  |
|  | T_ship_urb_coa |  | Coarse |  |
| C_SMOD_ad_mar | T_SMOD_mar_ait | SMOD | Aitken | Marine |
|  | T_SMOD_mar_acc |  | Accumulation |  |
|  | T_SMOD_mar_coa |  | Coarse |  |
| C_SMOD_ad_con | T_SMOD_con_ait | SMOD | Aitken | Clean Continental |
|  | T_SMOD_con_acc |  | Accumulation |  |
|  | T_SMOD_con_coa |  | Coarse |  |
| C_SMOD_ad_urb | T_SMOD_urb_ait | SMOD | Aitken | Urban |
|  | T_SMOD_urb_acc |  | Accumulation |  |
|  | T_SMOD_urb_coa |  | Coarse |  |

For this work, the Shipway activation scheme was modified to account for a temperature change due to both adiabatic and non-adiabatic processes, using Eq. (9). Aerosol loadings from Whitby (1978) were used to test both activation schemes. These properties considered different environments, ranging from clean to polluted (Table 1). The temperature was set as a fixed value of 274 K, based on surface temperatures observed during fog formation (Price, 2011; Haeffelin et al., 2013). All tests were driven by cooling rates found in fog formation (1 - 4 K hr$^{-1}$ calculated using data from Price, 2011), and also accounted for a temperature change due to a nocturnal clear sky cooling (0 - 1 K hr$^{-1}$; Kiehl and Trenberth, 1997).

Table's 2 and 3 displays the case setups used in the offline box model, including the list of tests conducted in each case. Table 2 lists all the tests that directly compared the Shipway and SMOD scheme, based on Eq.'s (4) and (9). Each scheme is tested with all combinations of the three different environments (marine, clean continental and urban) and the three different aerosol modes (Aitken, accumulation, coarse). To check that the SMOD scheme was correctly coded into the box model, meaning that supersaturation can be driven by a cooling rate rather than an updraft velocity, the non-adiabatic term in SMOD and $w_{min}$ was set to zero. Conducting this case would also test the aerosol activation's sensitivity to the choice in aerosol mode.

**Table 3.** The tests conducted in the offline box model used to test activation scheme representation and appropriate use of $w_{min}$.

| Case | Tests in case | Scheme used | Cooling source | $w_{min}$ applied | Aerosol mode | Environment |
|---|---|---|---|---|---|---|
| C_accumulation_mar | T_ship_mar_acc | Shipway | Adiabatic | | Accumulation | Marine |
| | T_ship_mar_acc_wmin | Shipway | Adiabatic | x | | |
| | T_SMOD_mar_acc | SMOD | Non-adiabatic | | | |
| C_accumulation_con | T_ship_con_acc | Shipway | Adiabatic | | Accumulation | Clean continental |
| | T_ship_con_acc_wmin | Shipway | Adiabatic | x | | |
| | T_SMOD_con_acc | SMOD | Non-adiabatic | | | |
| C_accumulation_urb | T_ship_urb_acc | Shipway | Adiabatic | | Accumulation | Urban |
| | T_ship_urb_acc_wmin | Shipway | Adiabatic | x | | |
| | T_SMOD_urb_acc | SMOD | Non-adiabatic | | | |

Further cases (details in Table 3) were run in order to identify the impact of $w_{min}$ in the Shipway scheme, and the impact of the non-adiabatic term in SMOD. For this case, three representations were used:

1. SMOD, which accounted for both adiabatic and non-adiabatic cooling. For these tests, $w_{min} = 0 \text{ m s}^{-1}$ and the adiabatic cooling component was switched off;

2. Default Shipway scheme. Cooling is assumed to be adiabatic, with $w_{min} = 0.1 \text{ m s}^{-1}$;

3. Shipway scheme, with cooling assumed to be adiabatic and $w_{min} = 0 \text{ m s}^{-1}$ (i.e. assuming no additional sub-grid cooling). This might be more appropriate for use in an LES where vertical motion is well resolved.

Firstly, a comparison between Shipway with no $w_{min}$ and Shipway in its default setting (with an applied $w_{min} = 0.1 \text{ m s}^{-1}$) tested the suitability of a $w_{min}$ in fog modelling. This comparison was motivated by Boutle et al. (2018), who discussed how aerosol activation in fog can be overestimated by the use of a $w_{min}$ value designed for convective clouds. The results of this test will quantify this overestimation, and hence guide how $w_{min}$ may require modification for fog modelling. Next, a comparison between SMOD and Shipway with $w_{min} = 0 \text{ m s}^{-1}$ tested the suitability of assuming adiabatic cooling in a non-adiabatic

environment.

## 4 Testing Shipway and SMOD using an offline box model

### 4.1 Behaviours of the Shipway and SMOD scheme in low updraft velocity regimes

This section's objective is to understand the relative importance of different aerosol modes concerning aerosol activation in fog and to check that the adiabatic pathway in the SMOD scheme was coded correctly. Although the implementation for SMOD

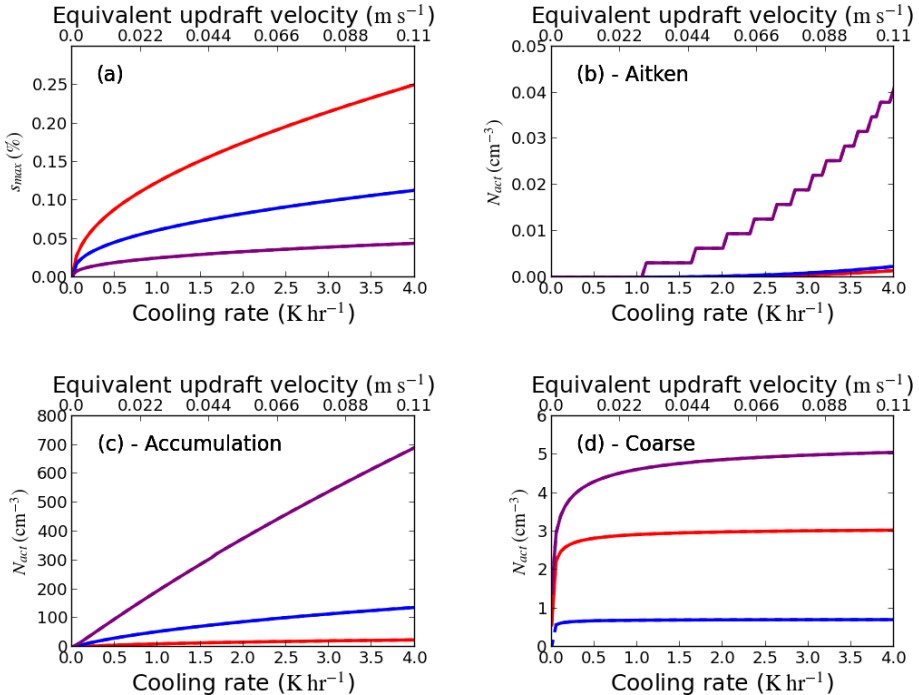

**Figure 1.** (a) Maximum supersaturation, $s_{\mathrm{max}}$ (%), against the total cooling rate. (b) - (d) A plot of activated aerosol concentration, $N_{\mathrm{act}}$ (cm$^{-3}$) against the total cooling rate for Aitken, accumulation and coarse mode aerosols respectively. Red - marine; Blue - clean continental; Purple - urban. Solid line - T_ship_ad; Dashed line - T_SMOD_ad (solid line overlaying the dashed line).

is different in that it applies a cooling rate rather than an updraft velocity, these tests comparing Shipway to SMOD should produce identical results for a given equivalent cooling rate.

When comparing the code that would control the adiabatic pathways in the Shipway and SMOD scheme, the differences in numerical calculations are negligible across all tests, which is shown by the overlapping dashed line over the solid line for all tests in Fig. 1. Figure 1a shows a monotonic increase in the maximum supersaturation, $s_{\mathrm{max}}$, across all environments with respect to updraft velocity. For a fair comparison, an equivalent cooling rate was calculated for the SMOD scheme using the dry adiabatic lapse rate assumption (see Eq. A6 in Appendix A). The $s_{\mathrm{max}}$ is 0.26% for the marine environment; corresponding to a cooling rate of 4 K hr$^{-1}$, and decreases as the aerosol concentration increases (0.11 and 0.04% for the clean continental and urban environment respectively). The decrease in $s_{\mathrm{max}}$ with increases in aerosol concentration relates to increased water vapour competition and hence condensation rate, resulting in a reduced likelihood of newly activated droplets.

Figures 1b-d show an increase in activated aerosols in relation to cooling rate. Of the three modes, the proportion of activated aerosols is greatest in the accumulation mode in all tested environments. This is even though in some environments (e.g. marine), the proportion of aerosol in the Aitken mode is greater than the accumulation mode (see Table 1). The relatively small radii of Aitken mode aerosol compared to the rest of the aerosol spectrum makes the required maximum supersaturation for activation significantly higher, as displayed in Fig. 1b. The reality is that supersaturation levels in fog have been shown

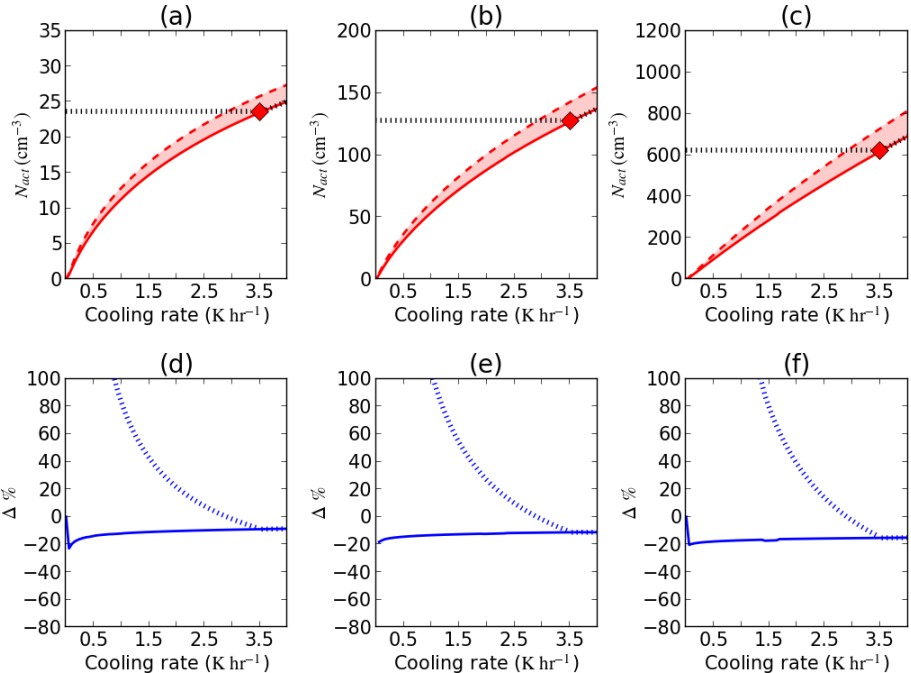

**Figure 2.** (a) Total activated aerosols, $N_{act}$, against the cooling rate for marine environment accumulation mode aerosols. Solid line - T_ship_mar_acc; dashed line - T_SMOD_mar_acc; black dashed line - T_ship_mar_acc_wmin. (d) Percentage differences, $\Delta$ %, between: dashed line - T_ship_mar_acc against T_ship_mar_acc_wmin; solid line - T_ship_mar_acc against T_SMOD_mar_acc. Red diamond - $w_{min} = 0.1 \mathrm{m\ s^{-1}}$ (b), (e): clean continental; (c), (f): urban.

to only reach several tenths of 1% (Gerber, 1991), and hence would not be great enough to activate Aitken mode aerosol. Given the result of this test, it could indicate that nocturnal fog simulations that account for aerosol activation can neglect the Aitken mode. This will be discussed further in Section 5. Although there is an increase in $N_{act}$ with respect to updraft velocity for Aitken mode aerosol (Fig. 1b), the aerosol activation fraction is so small that it leads to a visible stepwise function (this being strongest in the urban environment). The stepwise behaviour for the Aitken mode is a result of poor resolution in the

look-up table for the Shipway scheme at low updraft velocities, where this behaviour has been highlighted due to $w_{min}$ not being present. The resolution could be improved by using a more robust integration method. However, changing the integration method does not impact the general conclusions relating to the new scheme and hence this will be explored in later work.

### 4.2    Associated percentage difference for methods of aerosol activation

    To understand how $N_{act}$ may be impacted by the choice in aerosol activation representation, accumulation mode tests displayed

in Table 3 were rerun using the SMOD activation scheme and the Shipway (2015) scheme with an applied $w_{min}$. Although these same tests were run for Aitken and coarse mode aerosol (see Table 1), there was little to no change in $N_{act}$ when the aerosol activation representation was changed (not shown). In case C_accumulation_mar, T_SMOD_mar_acc produces a higher $N_{act}$

than T_ship_mar_acc for all cooling rates (Fig. 2a - marine), with a similar pattern being applicable to the clean continental and urban environments (Figs. 2b and c). As the SMOD scheme for these tests assumes non-adiabatic cooling exclusively, the increase in $N_{act}$ is due to the associated thermodynamical function being independent of adiabatic lifting and hence a change in pressure (see Appendix A for further details). Therefore, this demonstrates the dependency on the total number of activated aerosols on how the cooling is applied. To understand the impact of a $w_{min}$ threshold on $N_{act}$, all tests using the Shipway activation scheme were rerun, with the $w_{min}$ threshold of 0.1 m s$^{-1}$ being applied (Tests T_ship_mar_acc_wmin, T_ship_con_acc_wmin and T_ship_urb_acc_wmin). Applying this threshold resulted in a fixed $N_{act}$ for a cooling rate below 3.51 K hr$^{-1}$. Consequently, should there be a cooling rate lower than this threshold, $N_{act}$ will be overestimated and this may impact properties of the fog evolution such as the fog's optical depth.

Figures 2d, e and f show the percentage difference between the SMOD and Shipway (with an applied $w_{min}$) activation schemes increases as the prescribed cooling rate decreases. When comparing the three environments, the rate of increase in the percentage difference grows, as the tested environment becomes more polluted. For example, a cooling rate of 1.5 K hr$^{-1}$ results in a percentage difference of 40, 50 and 70% for the three environments respectively. Given the associated percentage difference, this indicates aerosol activation in fog simulations is overestimating $N_{act}$ by an appreciable amount. However, reducing the minimum threshold, $w_{min}$, to give an equivalent cooling rate close to those observed in fog would reduce but not remove the problem associated with the percentage difference. Between the SMOD and Shipway schemes for aerosols in the accumulation mode, the associated percentage change ranges between -10 and -20% for all three environments, and the rate of change in the percentage difference is not appreciably different for any given environment (Figs. 2d, e and f). This implies that even if the minimum threshold of $w_{min}$ were to be reduced such that it is representative for updraft velocities found in radiation fog, just using the Shipway scheme could potentially underestimate aerosol activation.

## 5  Testing Shipway and SMOD using MONC

The offline box model results demonstrate that assumptions widely used in aerosol activation (e.g. Abdul-Razzak and Ghan, 2000) may be significantly overestimating aerosol activation in fog. This section will investigate the impact that aerosol activation representation will have on fog evolution, using the Met Office Natural Environment Research Council Cloud (MONC) model (Brown et al., 2015, 2018). MONC is a large-eddy simulation model designed to research and develop parameterisations used in the forecast model. MONC and has the same equation set as the older Met Office Large Eddy Model (LEM; Gray et al., 2001) and unlike the LEM, MONC has been designed to couple with other modules, including the Cloud AeroSol Interactive Microphysics scheme (CASIM; Grosvenor et al., 2017; Miltenberger et al., 2018) and the Suite of Community Radiative Transfer codes (SOCRATES; Edwards and Slingo, 1996). MONC is widely used in the UK atmospheric science community, and has been used to study atmospheric processes in low level clouds in West Africa (Dearden et al., 2018), fog (Poku et al., 2019) and idealised convection simulations (Böing et al., 2019).

**Table 4.** The input parameters and model setup for IOP1 in MONC.

| IOP1 input parameters | Values |
|---|---|
| Horizontal domain | 132 x 132 m |
| Vertical domain | 705 m |
| $\Delta$x, $\Delta$y | 2 m |
| $\Delta$z | Variable - 1 m first 100 m, streched up to 6 m afterwards |
| Simulation duration | 12 hr |
| Timestep | 0.1 s |
| Surface geostrophic winds | $u_g$ = 1.3 m s$^{-1}$, $v_g$ = 2.1 m s$^{-1}$ |
| Cloud microphysics | Cloud AeroSol Interactive Microphysics (CASIM) |
| Radiative transfer scheme | Suite of Community RAdiative Transfer codes (SOCRATES) (Edwards and Slingo, 1996) |

## 5.1 MONC model - online setup

As part of this work, MONC is used to perform a suite of sensitivity tests based on intensive observation period 1 (IOP1) from the recent Local And Non-local Fog EXperiment (LANFEX) field campaign (Price et al., 2018). A full description of IOP1 and the observed vertical profiles the model was initialised with, can be found in Poku et al. (2019). The model setup for IOP1 is presented in Table 4. A domain size of $132 \times 132$ m$^2$ was chosen, as there is minimal impact on the fog's turbulent kinetic energy (TKE) and liquid water when compared to simulations that were tested on a larger domain (not shown). Although

previous studies such as Maalick et al. (2016) and Maronga and Bosveld (2017) have run LES fog simulations at higher horizontal resolutions, we found that running our cases at 2 m allowed for us to address our objectives, whilst compromising on both data storage and computational expense (not shown). The model's surface boundary conditions were prescribed with a varying surface temperature (described in Poku et al., 2019) and a surface vapour mixing ratio of 0.004 kg kg$^{-1}$, which were both based on observations. Radiation was calculated using SOCRATES based on the work of Edwards and Slingo (1996).

SOCRATES was called by the MONC model every 30 secs, allowing for the longwave radiative fluxes at the top of the fog layer to be captured in the model.

All simulations use the CASIM scheme; a multi-moment bulk microphysics scheme designed to simulate aerosol-cloud interactions (Grosvenor et al., 2017; Dearden et al., 2018; Miltenberger et al., 2018). For this work, CASIM has been set to 2 moments and is being used to represent a non-precipitating, warm boundary layer cloud (i.e. ice processes and autoconversion

to rain are turned off). In CASIM, the cloud-drop size distribution, $N(D)$, assumes a gamma distribution, which has the form (Shipway and Hill, 2012):

$$N(D) = N_0 D^{\mu_d} e^{-\lambda_d D}, \tag{11}$$

**Table 5.** Details of the simulations using the Shipway and SMOD activation scheme. The value of $w_{min}$ has been lowered from 0.1 to 0.01 m s$^{-1}$ based on the results from Sec. 4. Cooling rate equivalent calculated using the dry adiabatic lapse rate assumption.

| Test no. | Test name | Scheme | Imposed $w_{min}$ (m s$^{-1}$) | Threshold cooling rate equivalent (K hr$^{-1}$) | $r_e$ ($\mu$m) |
|---|---|---|---|---|---|
| T1 | T_shipway_wmin | Shipway | 0.1 | 3.51 | 10 |
| T2 | T_shipway_0.01 | Shipway | 0.01 | 0.351 | 10 |
| T3 | T_SMOD | SMOD | N/A | N/A | 10 |
| T4 | T_SMOD_er_15 | SMOD | N/A | N/A | 15 |
| T5 | T_SMOD_er_20 | SMOD | N/A | N/A | 20 |

where $N_0$ is the distribution intercept parameter, $\mu_d$ is the shape parameter (the default value of $\mu_d$ is set to 0), $\lambda_d$ is the slope parameter and $D$ is the droplet diameter. For this work, $\mu_d$ has been set to equal 3.0, based on observations of the liquid water path (LWP) and cloud-drop size distribution during IOP1, resulting in a more sensible modelled sedimentation rate (see Appendix B for details).

During IOP1, there were no direct aerosol or CCN measurements. Therefore, we initially planned to use a multi-mode log-normal aerosol distribution of 1000 cm$^{-3}$ Aitken-mode aerosols (mean diameter 0.05 $\mu$m), 100 cm$^{-3}$ accumulation-mode aerosols (mean diameter 0.15 $\mu$m) and 2 cm$^{-3}$ coarse-mode aerosols (mean diameter 1 $\mu$m), each following a standard deviation of 2.0, as proposed and used in Boutle et al. (2018). Using these values would therefore being representative of the clean air typically found at Cardington. However, our simulations used a single accumulation aerosol mode to maintain consistency with the tests in the Shipway Box Model, which showed that when considering aerosol activation, the activated $N_a$ for IOP1 can be accounted for by accumulation mode aerosols (not shown). A consequence of assuming a single accumulation mode potentially limits droplet concentration overestimation, which would lead to the fog layer transitioning too quickly in optical thickness. However, based on our offline test results, we believe that using a multi-mode aerosol spectrum would have led to an unnecessary computational expense in this study. This reasoning may be different should these simulations have been run with a prognostic for supersaturation, but this is outside the scope of this work. To reduce computational expense and data storage, 1D diagnostics are output every 1min and 3D diagnostics are output every 5min.

SMOD was implemented into MONC based on Eq. (9), which involves adding the adiabatic and non-adiabatic contributions together for the combined cooling rate to them be used for aerosol activation. The adiabatic contribution for this equation was derived from the resolved positive updraft velocity in MONC. The non-adiabatic contribution to date only consists of the longwave heating tendency that is derived using SOCRATES. For reference, the implementation of these partitioned terms is done similarly to the aerosol activation scheme used by Vie et al. (2016). Although it has been acknowledged that there are other non-adiabatic contributions to changes in supersaturation such as turbulent mixing, further model development would be required to account for these changes. However, given that radiative cooling is the biggest source of saturation during fog formation (Roach et al., 1976), these results should provide useful insight into the representation of aerosol activation during a stable fog case.

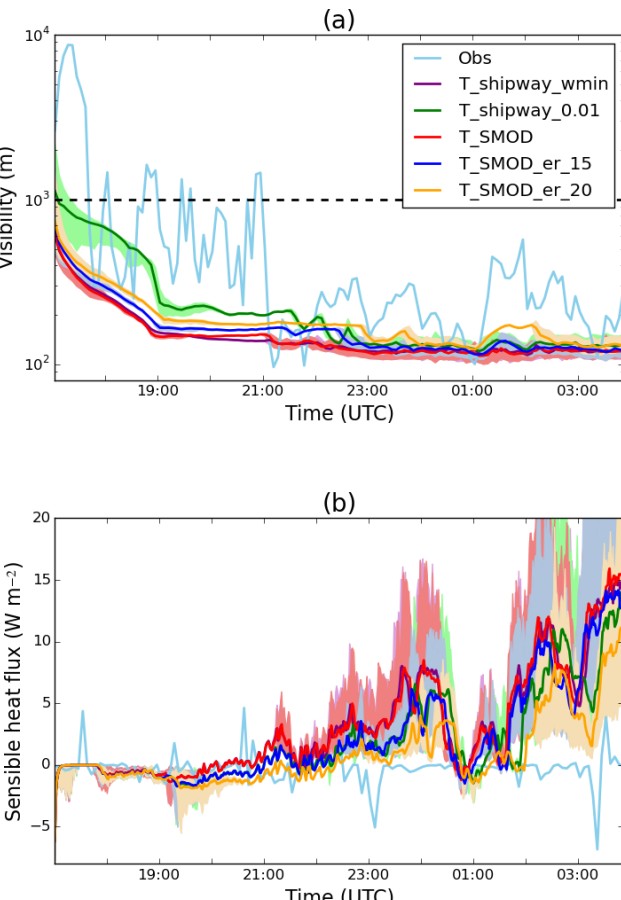

**Figure 3.** (a) - Time series of the near-surface mean visibility ($Vis$; m) at a 2 m altitude. Purple – T_shipway_wmin; green – T_shipway_0.01; red – T_SMOD; blue – T_SMOD_er_15; orange – T_SMOD_er_20; light blue – observations. (b) - Time series of the surface sensible heat flux (W m$^{-2}$). Purple – T_shipway_wmin; green – T_shipway_0.01; red – T_SMOD; blue – T_SMOD_er_15; orange – T_SMOD_er_20; light blue – observations. Minimum and maximum (a) visibility and (b) sensible heat flux are marked on the figure by the shaded area.

Table 5 displays all tests that will evaluate Shipway against SMOD. The first objective will be addressed by comparing test's T1 - T3 and the outcome of this comparison will improve the understanding of how different activation representations could influence the fog droplet number concentration (FDNC) evolution in IOP1. To date, the effective radius, $r_e$, has the option to be fixed or for it to vary with a change in FDNC. To isolate the impact of aerosol activation on number concentration, this work used a fixed $r_e$. As the non-adiabatic contribution in the SMOD scheme is directly influenced by $r_e$, two tests were setup testing its sensitivity, and hence will motivate future work that involves deciding whether a coupled effective radius is required when using the SMOD scheme.

## 5.2 Comparing simulations using the Shipway and SMOD scheme

Fog forms in tests T_shipway_wmin, T_shipway_0.01 and T_SMOD at 1700 UTC, and all decrease to a mean near-surface visibility of 120 m by the end of the night (Fig. 3a). For all model simulations, visibility, $Vis$, is calculated using the formula of Gultepe et al. (2006):

$$Vis = \frac{1.002}{(LWC \times FDNC)^{0.6473}}, \tag{12}$$

where $LWC$ is the liquid water content and $FDNC$ is the fog droplet number concentration. Equation (12) was derived based on observations of fog in mainland Europe and is valid over a range of droplet concentrations from a few per cubic centimetre up to a few hundred per cubic centimetre (Gultepe et al., 2006).

Despite the differences in near-surface visibility, all three tests have the strongest rate of decrease between 1700 and 1845 UTC (Fig. 3a). During this time, the mean near-surface visibility in T_shipway_wmin, T_shipway_0.01 and T_SMOD decrease to 208, 151 and 210 m respectively. T_shipway_0.01 has a noticeably higher near-surface visibility before 1830 UTC and best agrees with observations, before decreasing in visibility at the same rate as T_shipway_wmin. Upon first inspection, it appears that just lowering $w_{min}$ is the solution to prevent the simulation overestimating aerosol activation in fog, as shown by T_shipway_0.01. However, the model's spin-up period lasted around an hour in these simulations, meaning that the FDNC calculation is likely being influenced by initial prescribed random perturbations, as opposed to turbulence driven by either wind shear or convective motion. Unfortunately, with the earliest radiosonde data available being at 1700 UTC, the features in T_shipway_0.01 could not be avoided (for context, the observations show a stable boundary layer (SBL) beginning to form around the time of model initialisation). Nonetheless, the lower threshold used in T_shipway_0.01 allows for the simulation to undergo a slower transition in near-surface visibility to a thicker fog. This suggests that the number of activated droplets calculated may account for an inaccurate representation of what was observed during IOP1.

Up until 2100 UTC, T_shipway_wmin and T_SMOD and T_shipway_0.01 mostly experience a zero or slightly negative sensible heat flux (SHF), which agrees well with observations (Fig. 3b). After 2100 UTC, all three simulations grow positively in SHF, with both T_shipway_wmin and T_SMOD experiencing two distinct maxima of 6 and 14 W ms$^{-2}$ at times 0000 and 0400 UTC respectively. A mostly positive SHF is due to the fog layer growing enough in both depth and optical thickness that it will begin to warm the surface (Price, 2011). As the observed SHF was mostly zero throughout the night, our results indicate that the default $w_{min}$ used in the Shipway scheme will lead to the fog episode transitioning to a well-mixed layer too quickly. Up until 0100 UTC, T_shipway_0.01 has a lower SHF than T_shipway_wmin and T_SMOD, despite it remaining positive. This result highlights the inaccuracy of simulating fog cases with just an updraft velocity, providing further evidence for the use of the SMOD scheme. However, despite this suggestion, T_SMOD in its default settings is performing worse than T_shipway_0.01. This discrepancy will be discussed further in Section 5.2.1 of this paper. There is a possibility that the simulated SHF results may have some uncertainty due to no land-surface scheme being coupled to MONC to date, which has been shown to be important when the fog becomes optically thick (e.g. Porson et al., 2011). Unfortunately, investigating this uncertainty is outside the scope of this work.

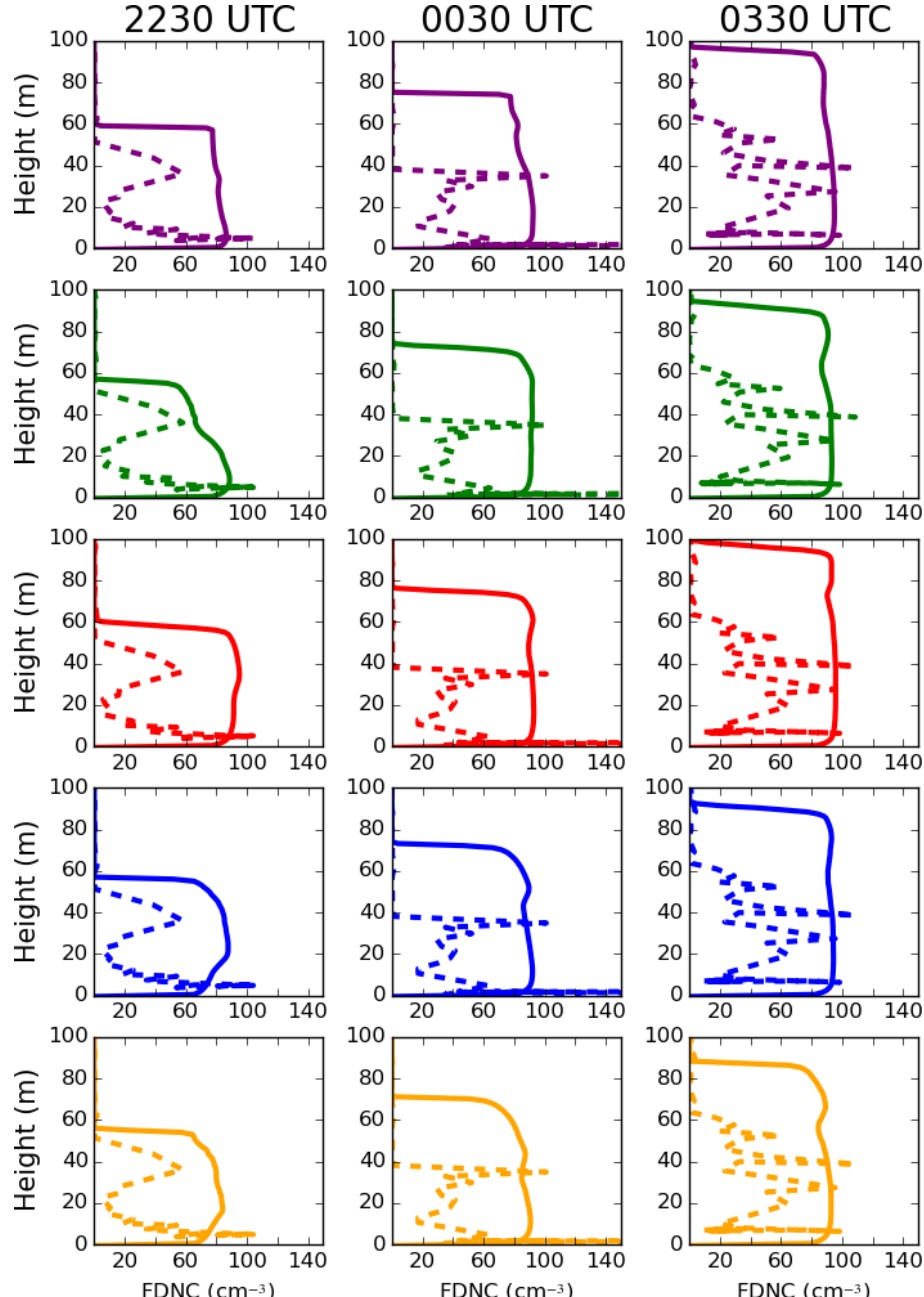

**Figure 4.** Vertical profiles of the fog droplet number concentration (cm$^{-3}$) at 2230, 0030 and 0330 UTC. The dashed lines represent observations, and solid lines represent simulated values. Purple – T_shipway_wmin; green – T_shipway_0.01; red – T_SMOD; blue – T_SMOD_er_15; orange – T_SMOD_er_20.

**Table 6.** Ratio of modelled to observed cloud drop number averaged over the vertical height across tested time frames (3 sf).

| Test name | 2230 UTC | 0030 UTC | 0330 UTC |
|---|---|---|---|
| T_shipway_wmin | 2.77 | 1.66 | 2.99 |
| T_shipway_0.01 | 2.54 | 1.69 | 2.96 |
| T_SMOD | 3.03 | 1.70 | 3.07 |
| T_SMOD_er_15 | 2.78 | 1.65 | 3.00 |
| T_SMOD_er_20 | 2.63 | 1.59 | 2.92 |

Vertical profiles of observed fog droplet number concentration (FDNC) initially show spatial variation throughout the layer, where it begins to homogenise throughout the night (Fig. 4). At 0030 UTC, it appears as though the fog layer decreased in height. However, this feature is most likely due to an instrumentation limitation, resulting in only accounting cloud droplets that were of sizes between 2 and 40 $\mu$m in diameter, with a 1 $\mu$m uncertainty (Price et al., 2018). Therefore, there is a potential that droplets that have begun growing through condensation were not accounted for. Within the fog layer, T_shipway_wmin and T_SMOD both have an activation rate between 75-80%, which increases to around 90% later in the night. Furthermore, the modelled to observed fog droplets for both simulations is 2.77 and 3.03 respectively (Table 6). Consequently, both the simulated activation and proportion rates lead to the fog layer growing 40 m too deep in comparison to observations. Initially, T_shipway_0.01 has the lowest droplet proportion rate, with the simulated spatial vertical FDNC agreeing best with observations. However, it begins to have activation and proportion rates similar to T_shipway_wmin and T_SMOD. This result suggests that T_shipway_0.01 transition rate to a thicker fog is still too fast, indicating that using an activation scheme driven by just an updraft velocity is unsuitable for fog simulations.

Throughout the night, both T_shipway_wmin and T_SMOD have a higher LWP than T_shipway_0.01 (Fig. 5). Poku et al. (2019) showed that the LWP increases with aerosol concentration and hence FDNC, with Porson et al. (2011) demonstrating that the increase in FDNC resulted in a stronger downwelling longwave flux, signalling the presence of a deeper fog. T_shipway_wmin has the steepest decrease in visibility during fog formation, suggesting that it has the highest initial FDNC. As these tests all have the same fixed effective radius (unlike studies such as Stolaki et al., 2015), the change in LWP is primarily due to the sedimentation rate, therefore indicating that T_shipway_wmin has the slowest sedimentation rate of all three tests as a result. A decreased sedimentation rate will lead to more liquid water being present in the fog layer. Consequently, this will lead to stronger cooling at the fog top (Poku et al., 2019), strengthening the feedback of increased liquid water production in the layer. This result provides further evidence of how the error in aerosol activation that utilises a $w_{min}$ of 0.1 m s$^{-1}$ impacts the fog, especially during the initial formation stage. The LWP and mean near-surface visibility are not appreciably different between T_shipway_wmin and T_SMOD (Fig. 5), suggesting the FDNC is very similar between the two. T_shipway_0.01 has the highest near-surface visibility between 1700 and 2300 UTC by up to 340 m, in addition to the lowest LWP by up to 4 g m$^{-2}$. Averaged time-height slices of FDNC and LWC were taken for all three tests, showing relatively small changes in the fog layer's FDNC between T_shipway_wmin and T_SMOD (not shown).

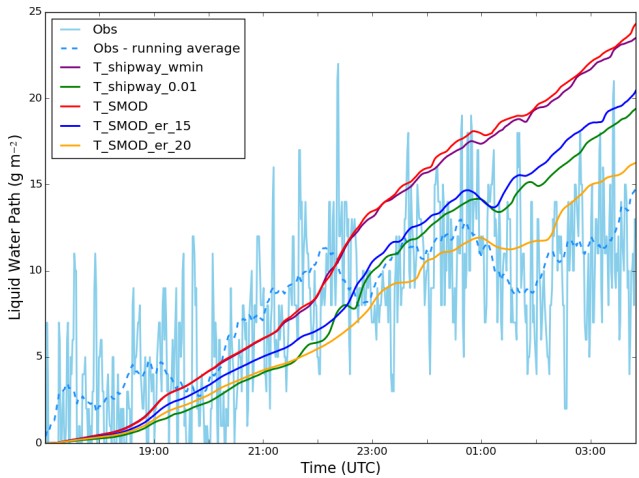

**Figure 5.** (a) - Time series of the surface deposition rate (g m$^{-2}$ hr$^{-1}$). Purple – T_shipway_wmin; green – T_shipway_0.01; red – T_SMOD; blue – T_SMOD_er_15; orange – T_SMOD_er_20; light blue – observations. (b) - Time series of the liquid water path (g m$^{-2}$). Purple – T_shipway_wmin; green – T_shipway_0.01; red – T_SMOD; blue – T_SMOD_er_15; orange – T_SMOD_er_20; light blue – observations; blue dashed - running average over observations (40 points).

So far we have seen that T_SMOD is performing similar to T_shipway_wmin, with T_shipway_0.01 appearing to be the
ideal solution for simulating IOP1. However, this may indicate that the default settings for SMOD may not be suitable for our study. The similarity of T_shipway_wmin and T_SMOD suggests that the combined cooling rate in T_SMOD is similar to the cooling rate associated with $w_{min}$ in T_shipway_wmin. To understand whether this is the case, a horizontal slice at $z = 2$ m of FDNC and the contributions to the relative cooling rates were taken at different times, as shown in Fig. 6. As 2 m is not at the model's lowest vertical grid box, there should not be any direct heating from the imposed surface conditions. At
1730 UTC the FDNC is about 83 cm$^{-3}$ (Fig. 6a), with more than 85% of the total cooling contribution being due to longwave heating (Fig. 6m). However, later in the night, the cooling contribution to longwave tendencies increases to around 90% within the fog layer (Fig. 6o), due to a decrease in the adiabatic cooling tendency to about 0.5 K hr$^{-1}$ (Fig. 6k). Eventually, the fog develops, resulting in the longwave contribution to cooling decreasing to around 15% (Fig. 6p), with an increase in cooling due to vertical motion. The drop in near-surface longwave cooling occurs as the fog transitions to become optically thick, and
so the longwave flux divergence becomes smaller near the surface, while the adiabatic effects become larger due to the onset of convection driven by radiative cooling at the fog top (Mazoyer et al., 2017).

The new SMOD activation scheme is more physically realistic, in that it is coupled to the radiative cooling in the fog, making the scheme potentially more sensitive to the way that this cooling is calculated in the model. Therefore, the assumption of the effective radius being fixed for these simulations may not be suitable to accurately simulate the radiative impact of the fog
layer. The following section will present some sensitivity tests to assess the impact of this assumption on fog development.

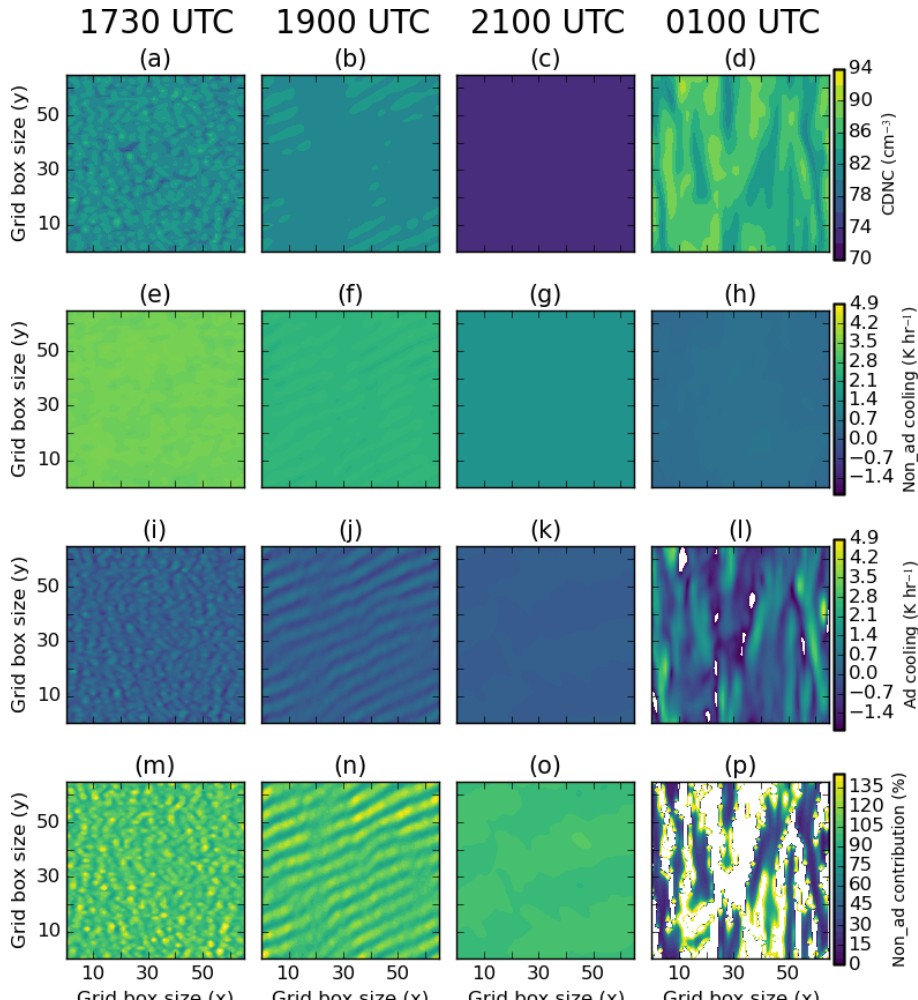

**Figure 6.** Horizontal slices made at z = 2 m of FDNC (cm$^{-3}$) in T_SMOD at (a) - 1730, (b) - 1900, (c) - 2100 and (d) - 0100 UTC. (e) - (h): non-adiabatic cooling (K hr$^{-1}$); (i) - (l): adiabatic cooling (K hr$^{-1}$); (m) - (p): Non-adiabatic cooling contribution (%). Note: for the cooling contribution, white masks out regions where the contribution is greater than 140 (%).

### 5.2.1   Sensitivity of SMOD to the effective radius

Hill et al. (2008) showed the impact of using a fixed effective radius on stratocumulus clouds simulations, with studies such as Bierwirth et al. (2013) and Young et al. (2016) showing how the observed effective radius in arctic clouds can change (between 5 to 15 $\mu$m) in relation to the cloud's LWC and FDNC. As this variability may be key to modelling radiation fog using the SMOD activation scheme, two tests were conducted that investigated the fog's sensitivity to a change in $r_e$.

When increasing $r_e$ from 10 to 20 $\mu$m, the near-surface visibility increases by up to 40%, and decreases the LWP by up to 42% (Fig. 3a). Furthermore, increasing $r_e$ leads to the simulated SHF better agreeing with observations, despite it still being


positive later in the night (Fig. 3b). However, although increasing $r_e$ results in the LWP agreeing better with observations (Fig. 5), neither test captures the changes in near-surface visibility during fog formation (Fig. 3b). Previous studies (e.g.
Bergot et al., 2015; Ducongé et al., 2020) argued that it's critical to account for a heterogeneous terrain when simulating the fog's initial spatial variability. However, Cardington is relatively homogeneous and hence potentially highlights a further discrepancy in the aerosol representation in these simulations. As an example, in-cloud removal (nucleation scavenging) has not been accounted for in this work, which has been shown to impact the spatial variability and development of mixed-phased clouds (Miltenberger et al., 2018). In addition, the discrepancy in our results may be due to these simulations utilising a bulk
microphysics scheme, which has been shown to not account for hydrated and small newly formed droplets (Schwenkel and Maronga, 2020). Nonetheless, the decrease in liquid water indicates that the fog's development in optical thickness has slowed down with an increase in $r_e$ and hence the importance of coupling both CASIM and SOCRATES together. Going forward, future studies should use a coupled $r_e$ as this should, in theory, lead to an improvement in FDNC as the better representation in aerosol activation in the SMOD scheme will feed into the radiation scheme.

Figure 7 shows time-height slices of FDNC and LWC for T_SMOD, T_SMOD_er_15 and T_SMOD_er_20. Before 2145 UTC, the FDNC in T_SMOD is strongest towards the top at around 80 cm$^{-3}$. After this time, it increases throughout the fog layer to a range between 86 and 94 cm$^{-3}$ (Fig. 7a). These changes in FDNC can be noted when compared to observations, where both T_SMOD_er_15 and T_SMOD_er_20 begins to have a less uniform structure throughout the fog layer (Fig. 4). Coinciding with this is an increase in LWC from 0.2 to 0.24 g kg$^{-1}$, suggesting the time at which the fog began to develop and
grow in optical thickness. However, an increase in $r_e$ results in delayed onset of the growth in optical thickness to 2300 and 0030 UTC for T_SMOD_er_15 and T_SMOD_er_20 respectively. The FDNC on average decreases for both T_SMOD_er_15 and T_SMOD_er_20 across the whole fog layer, with a noticeable rise at around 2300 UTC for T_SMOD_er_15. Although this pattern is the same for T_SMOD_er_20, there are periods where there are visible decreases in FDNC, e.g. between 0130 and 0230 UTC (Figs. 7c and e respectively). A combination of both the FDNC and LWC decreasing results in a slower transition
in the fog layer, which is shown in the downwelling longwave at 2 m (Fig.8). The downwelling longwave decreases by a maximum of 20 W m$^{-2}$ between T_SMOD and T_SMOD_er_20, with T_SMOD_er_20 undergoing the slowest positive rate with all the simulations presented in this paper. There are differences between the observed and simulated downwelling in all three simulations, however, before 2200 UTC, T_SMOD_er_20 decreases this difference to a maximum of 10 W m$^{-2}$.

SOCRATES calculates the longwave radiative fluxes by the cloud's optical depth, $\tau$, (Edwards and Slingo, 1996):

$$\tau = k^{(e)} \Delta m, \tag{13}$$

such that $\Delta m$ is the change in mass for a given spectral band and $k^{(e)}$ is the mass extinction coefficient, which is defined as:

$$k^{(e)} = L \left( a + \frac{b}{r_e} \right). \tag{14}$$

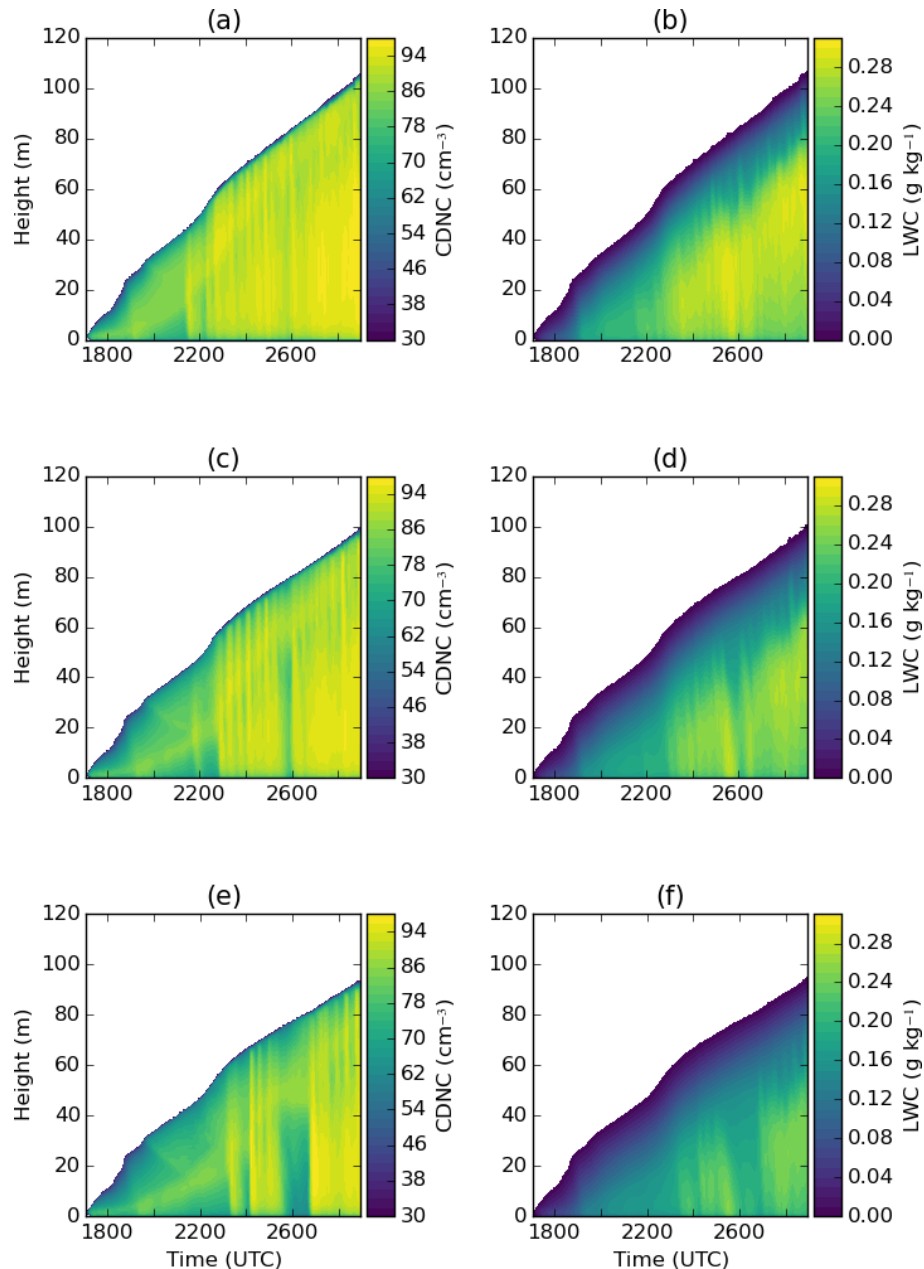

**Figure 7.** Plots of (a), (c), (e) - mean FDNC (cm$^{-3}$); and (b), (d), (f) - mean LWC (g kg$^{-1}$). (a), (b): T_SMOD; (c), (d): T_SMOD_er_15; (e), (f); T_SMOD_er_20.

For SMOD scheme, both the FDNC and LWC is sensitive to $r_e$, given Eq's (13) and (14). This leads to a more physical representation of aerosol activation that should be considered when simulating cases of fog. These results demonstrate the importance of an accurate effective radius and the reasons for using a coupled $r_e$, given its impact on the fog evolution.


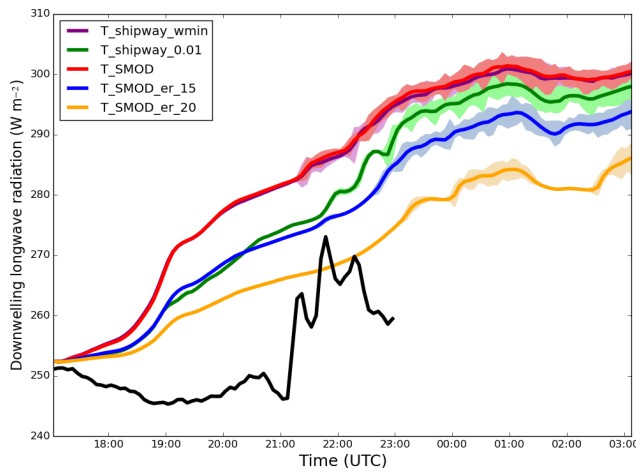

**Figure 8.** Time series of the downwelling longwave radiation (W m$^{-2}$) at a 2 m altitude. Purple – T_shipway_wmin; green – T_shipway_0.01; red – T_SMOD; blue – T_SMOD_er_15; orange – T_SMOD_er_20; black – observations. The minimum and maximum downwelling longwave radiation are marked on the figure by the shaded area.

## 6    Discussion

This work aimed to investigate how the representation of aerosol activation influenced nocturnal radiation fog simulations. There was a strong focus on critiquing the assumptions used in several current aerosol activation schemes, which are usually designed for clouds where cooling is driven by adiabatic ascent. This work addressed two research questions.

### 410    6.1    What are the potential differences in aerosol activation between the Shipway and SMOD scheme?

The assumptions used in the Shipway (2015) scheme to date, i.e. the use of just an updraft velocity with a minimum threshold $w_{min}$, were tested against the SMOD scheme in an offline box model. The sensitivity of Shipway to $w_{min}$ was first tested. For accumulation and coarse mode aerosol, there was a monotonical decrease in $N_{act}$ as $w_{min}$ approached 0 m s$^{-1}$. These tests also highlighted the stepwise function present in Aitken mode aerosol in the low updraft velocity regime. Given the fraction
of Aitken aerosols activated, our results may suggest that Aitken mode aerosol can be ignored when modelling activation in fog based on the range of environmental aerosol size distributions, as the required environmental supersaturation for impact is substantially higher than supersaturation's seen in reality. However, the stepwise behaviour was caused by the poor resolution in the look-up table that calculated $s_{max}$ in this regime, therefore demonstrating why just removing $w_{min}$ with no alternative cooling source may not be an appropriate solution when simulating aerosol activation in fog.
For accumulation mode aerosol, there were noticeable percentage differences between the actual cooling rate and the use of a $w_{min}$ equal to 0.1 m s$^{-1}$ (as typically used in clouds) by up to 70%, as the environment becomes more polluted. In reality, for a given liquid water path, increasing the aerosol concentration will result in a larger concentration of smaller droplets, increasing the fog's optical depth (Twomey, 1977), and may cause the fog to become well-mixed too quickly. Therefore for this example, a

similar effect could occur should an unsuitable $w_{min}$ be used in fog simulations. In addition, these tests demonstrated that using an aerosol activation scheme that assumes just adiabatic ascent may potentially underestimate $N_{act}$ by 20% in an environment driven by non-adiabatic cooling processes (i.e. fog formation). Furthermore, the associated percentage difference in the choice of $w_{min}$ would be the same should SMOD be run with just an adiabatic cooling source, given there were no differences in Shipway and SMOD in an adiabatic setup. Consequently, both of these results show that the aerosol indirect effects may not be properly accounted for in fog simulations when using a traditional aerosol activation scheme.

## 6.2 How do the differences in aerosol activation representation impact the fog evolution in a large eddy simulation?

The Shipway (2015) aerosol activation scheme was used to test the impact $w_{min}$ could have on simulating fog in MONC using only accumulation mode aerosol. It was shown that a reduction in $w_{min}$ lowered the initial FDNC during formation, resulting in the fog undergoing a slower transition to a well-mixed layer. Reducing $w_{min}$ to 0.01 m s$^{-1}$ displayed some unusual model behaviours during fog formation, which is most likely driven by the model's spin-up period, rather than shear or convection motion. However, the only way to confirm this is to initialise the model earlier, which is not possible with the given radiosonde data from IOP1. Upon initial analysis, there was not an improved performance using the SMOD scheme against the Shipway scheme with an applied $w_{min}$ of 0.1 m s$^{-1}$. Upon further inspection, it was shown that the cause of this result was due to $r_e$ not reflecting the change in FDNC. When $r_e$ was increased from 10 to 20 $\mu$m, the result was a slower transition to a well-mixed layer, which was more in line with observations of IOP1. This highlighted the importance of the effective radius and provides further motivation to couple the effective radius with a change in FDNC. However, despite this result using the SMOD scheme, our work has highlighted potential physical processes regarding aerosol missing in this study, demonstrating the complexities when simulating aerosol-fog interactions in nocturnal radiation fog.

## 7 Conclusions

This work has demonstrated the unsuitability of using an aerosol activation scheme designed for convective clouds in fog simulations, complementing previous studies such as Schwenkel and Maronga (2019), who have shown how the choice in aerosol activation scheme impacts the fog evolution through a change in the FDNC. Similar to our study, they and authors such as Mazoyer et al. (2017) have used a similar mathematical framework with their choice in using a non-adiabatic cooling in aerosol activation scheme first utilised by Zhang et al. (2014). Our work in this paper complements this and other studies by doing the following:

1. Although as suggested by Boutle et al. (2018) that the solution is to remove the $w_{min}$ threshold when simulating radiation fog, our results show that this is not necessarily a suitable option. This is highlighted with T_shipway_0.01, which although did initially perform better than the rest of the tests discussed, it transitioned quicker to a thicker fog than both T_SMOD_er_15 and T_SMOD_er_20. As aerosol activation in T_shipway_0.01 was driven by just an updraft velocity, this suggests why a more physical based activation scheme such as SMOD is critical to simulate nocturnal radiation fog. More specifically, the choice in activation scheme is key when the fog layer may transition to an optically thick layer.

2. Although there have been studies investigating the use of the non-adiabatic framework in fog simulations, this is the first study to the author's knowledge that critiques this framework with a fog case that formed in clean aerosol regimes (accumulation CCN $< 100 \, \text{cm}^{-3}$ as defined by Boutle et al., 2018), therefore supporting previous literature on the topic of aerosol-fog interactions.

460 Work to develop the SMOD scheme is still ongoing and will include a total non-adiabatic cooling tendency that will account for additional non-adiabatic processes such as turbulent or subgrid mixing. Completing this work could make it easier to incorporate the SMOD scheme into a model such as an NWP model. This is because the non-adiabatic process would be a change in temperature within the grid box, rather than requiring an explicit additional term. It was shown that SMOD is sensitive to SOCRATES with regards to the fixed effective radius, especially when considering the decrease in FDNC. Therefore, future 465 work should run the new scheme with the interactive coupling of $r_e$ to CASIM, should the option be available.

As noted in Section 5.2, SMOD was unable to capture the fog's spatial variability during initial formation. Poku et al. (2019) discussed how using a more "realistic" activation scheme such as SMOD would be a suitable solution, as the FDNC would be able to capture the fog's transitional period. However, although our work has shown that SMOD may be a better option than Shipway, our simulations were not able to simulate both the initial fog formation variability and remain stable throughout 470 the night. This feature may have been due to our study using a bulk microphysics scheme, and more specifically a saturation adjustment condensation calculation, which has shown to be problematic in other cloud regimes e.g. deep convective clouds (Lebo et al., 2012) and stratocumulus clouds (Thouron et al., 2012). Previous LES studies of IOP1 (e.g. Boutle et al., 2018) have addressed this limitation by using a prognostic for supersaturation, which in their works led to a reasonable transition to when the fog became optically thick. In addition, Schwenkel and Maronga (2020) proposed moving away from bulk microphysics 475 schemes and instead uses a Lagrangian cloud model (LCM), which can account for small droplets and swollen aerosols (for context, our results do not capture changes in the drop-size distribution with aerosol activation representation). Therefore, future work should include simulating IOP1 using an LCM, which could be capable to improve capturing the features of a thin fog.

Since our study was motivated to develop and test a suitable scheme that could be used in NWPs to account for aerosol 480 impacts in fog, an LCM mechanism may be unsuitable in an NWP due to additional computational expense. Furthermore, using a prognostic for supersaturation is unsuitable due to the timestep for changes in supersaturation being too small for most NWPs (Morrison and Gettelman, 2008). Miltenberger et al. (2018) showed that by including in-cloud aerosol removal, the source of aerosol began depleting through nucleation, resulting in a more open-cell cloud structure and changes in the cloud dynamics. As this study was done using a bulk microphysics scheme, this may be a suitable option when testing the SMOD 485 scheme. To date, there are no studies that have investigated the use of a nucleation scavenging parameterisation in fog in the context of bulk microphysical parameterisations, therefore suggesting a future piece of work within the subject of aerosol-fog interactions. For this work, there was a lack of simultaneous measurements of observed aerosol and cloud droplets. Given the $w_{min}$'s sensitivity to aerosol concentration, having these measurements in future studies will both help constrain the model and highlight any further discrepancies in aerosol activation representation in fog. Finally, our study has focused on the first 10 490 hours of IOP1 and hence has not accounted for the fog evolution during daytime. Given the impact additional processes such

as aerosol-radiation interactions and an interactive surface scheme will have on fog dissipation, it's critical to ensure schemes such as SOCRATES and CASIM are coupled for this future work.

As a wider implication, aerosol-cloud interactions are a big source of uncertainty when modelling atmospheric processes, both within forecasting (NWP) and climate (GCM) models and the choice of aerosol activation can influence how big this uncertainty is. Typically, the resolution of NWP and GCM model simulations is very coarse compared to LES, meaning that any present updraft velocities are usually subgrid and hence cannot be resolved. To represent aerosol activation on a subgrid level, the vertical velocity is either in the form a characteristic vertical velocity (e.g. Ghan et al., 1997) or a PDF function based on the vertical velocity (e.g. West et al., 2014). More recently, Malavelle et al. (2014), for example, discussed methods to account for subgrid velocities used in aerosol activation in convection-permitting models. These methods utilise a $w_{min}$, however, this should be lowered systematically for future work regarding aerosol activation in fog. Although gaining measurements of vertical velocity PDFs could be difficult in fog, the results presented in this paper could provide a useful framework to estimate what the variation in vertical velocities in fog could be, therefore providing a good estimation of the types of distributions that best match these velocities. Finally, to have a full cooling term applied in an NWP model, it is important to know how these vertical velocities correlate with the changes in non-adiabatic cooling.

This paper has shown the need to differentiate between optically thin fog ($w_{min} \approx 0$ m s$^{-1}$) and optically thick fog, where sub-grid vertical velocities can be important. The method being presented in this work is computationally efficient and provided an additional level of flexibility to consider different cooling sources in cases where updrafts are not the dominant cooling source. Given this flexibility, this will allow the SMOD scheme to undergo further testing in both high resolution and NWP models. Whilst this has been tested in only the Shipway and SMOD activation schemes, the framework for a change in supersaturation is generic enough for it to be applied to other activation schemes too.

## Appendix A: Mathematical formulation for the change in supersaturation

Pruppacher and Klett (2010) defined supersaturation in terms of the water vapour mixing ratio, $q_v$, as:

$$q_v = (1+s) \left( \frac{\epsilon e_s}{p} \right),\tag{A1}$$

where $p$ is the pressure of dry air, $s$ is the environment's supersaturation, $e_s$ is the saturation vapour pressure and $\epsilon = \dfrac{R_a}{R_v} = 0.622$; the ratio of the gas constant of dry air to water vapour. Differentiating Eq. (A1) with respect to time, and rearranging for the change in supersaturation gives:

$$\frac{ds}{dt} = \left( \frac{p}{\epsilon e_s} \right) \frac{dq_v}{dt} - (1+s) \left[ \frac{1}{e_s} \frac{de_s}{dt} - \frac{1}{p} \frac{dp}{dt} \right].\tag{A2}$$

The Clausius-Clapeyron equation is defined as:

$$\frac{de_s}{dT} = \frac{Le_s}{R_v T^2},$$
(A3)

with $L$ being defined as specific latent heat. Applying the chain rule gives:

$$\frac{de_s}{dt} = \frac{Le_s}{R_v T^2} \left. \frac{dT}{dt} \right|_{tot}.$$

$$= \frac{Le_s}{R_v T^2} \left[ \left. \frac{dT}{dt} \right|_{ad} + \left. \frac{dT}{dt} \right|_{non\_ad} + \left. \frac{dT}{dt} \right|_{lat} \right].$$
(A4)

$\left. \frac{dT}{dt} \right|_{ad}$ is the change in temperature due to dry adiabatic processes, such that:

$$\left. \frac{dT}{dt} \right|_{ad} \equiv -\Gamma \frac{dz}{dt} = -\Gamma w;$$
(A5)

where $\Gamma = \frac{g}{c_p}$, the dry adiabatic lapse rate with $c_p$ being the specific heat capacity, and $w$ is the updraft velocity. $\left. \frac{dT}{dt} \right|_{non\_ad}$ is the change in temperature due to non-adiabatic processes (e.g. radiative cooling, turbulent mixing), that excludes latent heat release, and $\left. \frac{dT}{dt} \right|_{lat}$ is the change in temperature due to latent heat release i.e. condensation/evaporation. For adiabatic expansion (lifting), there are corresponding pressure and temperature changes (that satisfy the first law of thermodynamics). However, for isobaric non-adiabatic heating processes, there is no change in $p$ but there is a change in $T$ that modifies Eq. (A4). Therefore, for the change in $p$, by:

1. assuming hydrostatic equilibrium, where $\frac{dp}{dz} = -\rho g$;

2. using the equation for the ideal gas law, where $p = \rho R_a T$;

$$\frac{dp}{dt} = \frac{dp}{dz} \frac{dz}{dt}$$

$$= -\frac{pg}{R_a T} w.$$
(A6)

The change in temperature due to latent heat release is proportional to the change in vapour mixing ratio, such that:

$$\left. \frac{dT}{dt} \right|_{lat} = -\frac{L}{c_p} \left. \frac{dq_v}{dt} \right|_{cond} = \frac{L}{c_p} \frac{dq_l}{dt}.$$
(A7)

Inserting Eq.'s (A4), (A6) and (A7) into Eq. (A2), and assuming $1 + s \approx 1$ gives:

$$\frac{ds}{dt} = \left( \frac{Lg}{R_v T^2 c_p} - \frac{g}{R_a T} \right) w - \frac{L}{R_v T^2} \left. \frac{dT}{dt} \right|_{non\_ad} - \left( \frac{p}{\epsilon e_s} + \frac{L^2}{R_v c_p T^2} \right) \frac{dq_l}{dt}. \tag{A8}$$

Eq. (A8) can be used to simulate aerosol activation in both fog and convective cloud regimes, highlighting the flexibility of the SMOD scheme. As an objective for this work is to understand how using an adiabatic framework to represent aerosol activation in an non-adiabatic environment (e.g. fog) may impact $N_{act}$, $w$ in Eq.(A8) will be rewritten as $\left. \frac{dT}{dt} \right|_{ad}$, such that:

$$\frac{ds}{dt} = \psi_1 \left. \frac{dT}{dt} \right|_{ad} + \psi_2 \left. \frac{dT}{dt} \right|_{non\_ad} - \gamma \frac{dq_l}{dt}, \tag{A9}$$

where:

$$\psi_1 = \frac{c_p}{R_a T} - \frac{L}{R_v T^2},$$

$$\psi_2 = -\frac{L}{R_v T^2}, \tag{A10}$$

$$\gamma = \frac{p}{\epsilon e_s} + \frac{L^2}{R_v c_p T^2}.$$

**Appendix B: Fitting modelled LWP and cloud drop-size distribution to observations - shape parameter**

All tests in this paper assume a fixed $r_e$, implying that the change in liquid water is controlled by the sedimentation rate, as
discussed in Poku et al. (2019). The sedimentation rate is controlled by the cloud drop-size distribution (see Eq. 11), with its skewness being determined by the shape parameter, $\mu_d$. Mazoyer et al. (2017) adapted the default shape parameter to best fit the modelled cloud drop-size distribution to observations. For this work, a similar approach would have ideally been chosen to find a suitable $\mu_d$ to capture the changes in liquid water. However, the instrumentation only began to record spectra during IOP1 4 hours into the observed fog case, and by this time, the layer had already begun to grow in optical thickness. To account for
this limitation, the LWP was used to decide on a suitable choice of $\mu_d$. These simulations were then compared to the available IOP1 cloud spectra data, to validate and hence choose a $\mu_d$ going forward. For this fitting, $\mu_d$ ranged from 0 to 3, with these tests denoted as T_mu_0, T_mu_1, T_mu_2 and T_mu_3 respectively. Although simulations were conducted to increase the shape parameter up to a value of $\mu_d = 7$ (similar to Mazoyer et al., 2017), the LWP for tests where $\mu_d > 4$ were higher than the observed mean LWP and hence these results will not be shown.
Both the surface deposition rate and LWP increase with $\mu_d$ (Figs. B1a-b), with this increase being more inline with observations. T_mu_2 and T_mu_3 both show improved LWP when compared to observations, especially before 2200 UTC. As there are potentially multiple options in choosing $\mu_d$, the modelled cloud drop-size distribution was compared to observations,

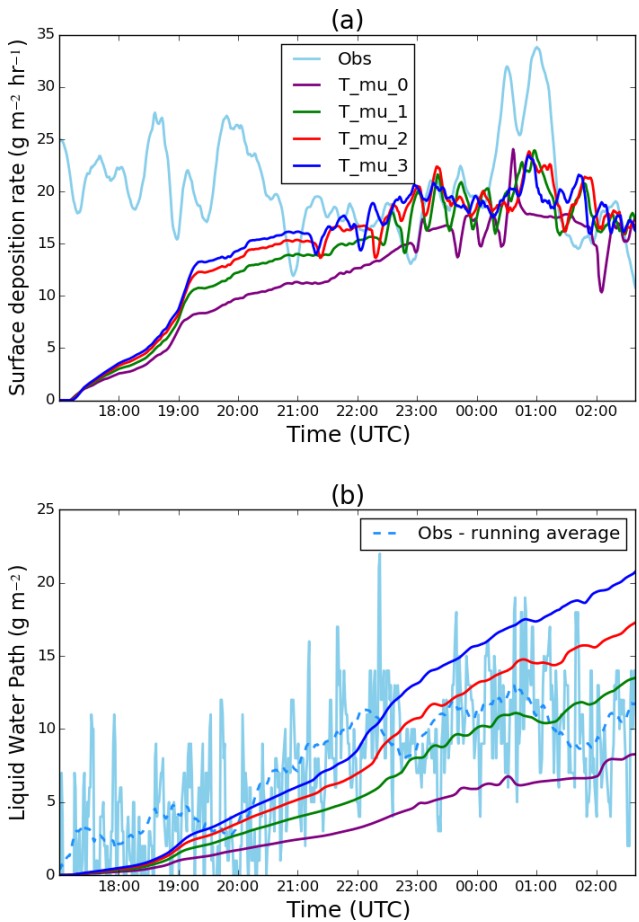

**Figure B1.** a) - Time series of the surface deposition rate (g m$^{-2}$ hr$^{-1}$). Purple – T_mu_0; green – T_mu_0; red – T_mu_1; dark blue – T_mu_3; light blue – observations. (b) Time series of the liquid water path (g m$^{-2}$). Purple – T_control; green – T_mu_1; red – T_mu_2; dark blue – T_mu_3; light blue – observations; blue dashed – running average over observations (40 points).

as shown in Fig. B2. Before 2200 UTC, all shape parameter tests began with an abundance of small droplets, signalling the formation of fog, and the density of small droplets being greatest in T_mu_0 (not shown). During fog evolution, all tests begin
560 moving right in terms of skewness with the exception for T_mu_0 (due to T_mu_0 being logarithmic). For the tests where $\mu_d > 0$, increasing the shape parameter results in the peak of the distribution decreasing and moving to the right, for all tested time frames. For example, increasing the shape parameter to $\mu_d = 3$ results in a peak droplet diameter of 11 $\mu$m. These results suggest a limitation in the default choice in $\mu_d = 0$ and hence the assumption of a logarithmic distribution for fog development during IOP1. By increasing the shape parameter during the fog evolution, fewer large droplets will sediment out of the fog
565 layer, therefore explaining the presence of bigger droplets still within the system in these tests (for example, tests T_mu_1 - 3).

  At 2200 UTC, the observed cloud droplet spectrum mostly follows a logarithmic distribution, however, later in the night, it evolves more into a bi-modal distribution (as seen in Price, 2011). For example, at 0000 UTC, the peaks occur at 8 and 22 $\mu$m.

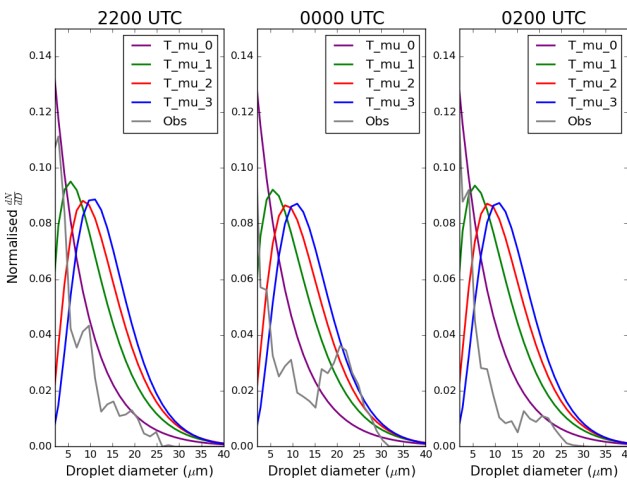

**Figure B2.** Cloud drop-size distributions for shape parameter simulations at 1710, 1800 and 2200 UTC at 2 m. T_mu_0; green – T_mu_0; red – T_mu_1; dark blue – T_mu_3; grey - observations.

Of the shape parameter tests, the observations are in best agreement with T_mu_3 for droplet size diameters between 22 to 27 $\mu$m at 0000 UTC, however, this fit does not take into account the peak shown within the smaller droplets. In an ideal situation,
a modelled cloud drop-size distribution would take into account the bi-modal nature shown within the distribution. In reality, it is likely that these smaller droplets have not activated, but instead are a source of hydrated aerosol which can contribute up to 68% of the total light scattered, and hence result in the reduction in visibility within the fog (Hammer et al., 2014). However, although these smaller droplets may potentially change the microphysical structure of the fog, the introduction of a bi-modal distribution (or a varying shape parameter) within CASIM may increase model computational expense, with no appreciable
changes in the fog evolution. Given these results, a shape parameter of $\mu_d = 3$ will be used in this paper.

*Author contributions.* CP undertook the research, carried out the simulations and analysis, and led the writing of the paper. AR, AB, AH contributed to the research design, interpretation of the results and the writing of the final paper. AH provided technical support with MONC. BS provided the box model and supported its use.

*Code availability.* The MONC, CASIM, offline box model and SOCRATES codes are maintained by the Met Office and accessible via the
Met Office Science Repository Service (https://code.metoffice.gov.uk/) (last access: 1st March 20210). The MONC branch is available at https://code.metoffice.gov.uk/svn/monc/main/branches/dev/craigpoku/r6496_MONC_poku_activation (last access: 28th August 2020). The CASIM branch is available at https://code.metoffice.gov.uk/svn/monc/casim/branches/dev/craigpoku/vn0.3.2_poku_scheme_debugged (last access: 1st March 2021). The offline box model branch is available at https://code.metoffice.gov.uk/svn/monc/casim/branches/dev/craigpoku/ r287_Offline_Activation (last access: 1st March 2021). For further details, please contact Adrian Hill (adrian.hill@metoffice.gov.uk)

*Competing interests.*   The authors declare that there are no competing interests.

*Acknowledgements.*   The authors would like to thank the Met Office Research Unit in Cardington (UK) for providing and processing the observational dataset used throughout this study. CP was supported by a Natural Environment Research Council (NERC) Industrial CASE award with the Met Office (grant number NE/M009955/1). This work used Monsoon2, a collaborative High Performance Computing facility funded by the Met Office and the Natural Environment Research Council. The authors would like to thank the anonymous reviewers for
comments that helped improve the quality of this manuscript.

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
