# Peer review of "Is a more physical representation of aerosol activation needed for simulations of fog?"

_Atmospheric Chemistry and Physics, 2020_

## Referee Comment (RC1) · Anonymous Referee #1 · 4 Oct 2020

The authors investigate the effect of different aerosol activation schemes on the simulation of radiation fog. In this course the present a modified scheme that is better suited for fog than the available ones, basically because these were designed for clouds in which adiabatic cooling is the main source for cooling. In fog, however, radiative cooling is the major process leading to cooling and aerosol activation. Past simulation studies on radiation fog have shown that the simulated fog layers usually deepened too rapidly. The authors are trying to demonstrate that this is due to deficiencies in the activation parameterizations. I thus believe the value of this work is high and can be an important contribution to fog research; and it is well suited to be published in ACP.

I do have a number of issues and comments the authors should address in their revised manuscript. I suppose they can be considered to be overall minor, but one issue (see

below) is major.

Detailed comments:

1. line 13-14: the sentence reads odd. The minimum updraft velocity threshold over-predicts the droplet number in comparison to a cooling rate? What you are saying is that a cooling rate underpredicts a droplet number. How can a cooling rate/threshold predict something?

2. Introduction: You elaborate the current state of fog simulation in which often a bulk microphysics scheme is used, nowadays often with aerosol activation parameterization. However, you have missed important recent work that avoids aerosol activation parameterizations by using a Lagrangian cloud model (see Schwenkel & Maronga 202, https://www.mdpi.com/2073-4433/11/5/466). I do think it would be worth to discuss their paper in the introduction. How do your results relate to their LES? Would a direct comparison of the same case make sense? The same authors also recently published a related article in ACP (https://acp.copernicus.org/articles/19/7165/2019/acp-19-7165-2019.html). You do cite this close to the end of the manuscript, but I would say their work is so closely related to what you did that I would expect that you put your work into context of the two papers in the introduction.

3. Eq 6: What is the variable "L"? I think you do not define it at all.

4. Table 2 / case definition. To be honest, i had serious difficulties to follow your text at some places because the case definition appears a little chaotic. The is best seen in Table 2. Example: Case C_adiabatic has two tests, but for T_sip_ad has all aerosol modes and all environments. So how many simulations were performed? It is very difficult to read the table that way. It would be beneficial to list all simulations in a more straight forward order.

5. In the legend of Figure 1 you refer to "Aitken, accumulation, and coarse model aerosols". However, if you are not an expert in aerosol physics you might not know

what these modes are. And this is the first time these terms appear in the manuscript! Please provide an explanation to the reader at a suitable location. It might help to give a schematic diagram showing these modes.

6. line 224: How can you motivate a grid spacing without given the grid spacing? Furthermore, you cite studies that clearly state that 1 m grid spacings is best choice, but you use only 2 m. Given the very limited horizontal domain, I have to ask: why did you use such "coarse" grid spacing? It does not make sense to me.

7. This is one of my major concerns. Your paper is supposed to illustrate that the SMOD scheme gives a smoother/delayed transition to deep fog and thus is more realistic. But what is almost completely missing is evidence for this. First of all, you show the LANFEX observations, but you do not really compare your LES results against them. And what I see is actually, that all runs performed are way off from the observations. So how can you conclude that the SMOD scheme is more realistic? It appears a rather academic idea that is not proven by better agreement with observations for instance (you see rather bad agreement e.g in Figures 3, 5, and 7). Second, where can we see this smoother transition to deep fog? The longwave downwelling radiation is a good indicator. Looking at Figure 7, however, it looks like the fog layer in general remains less thick, no matter what scheme is used. The transitions, however, happen at the same time. And even worse: in the LES all runs go into deep fog mode quickly, while the observations indicate shallow fog or no fog until 21:00, and then a rapid thickening. I would say the Shipway scheme here is the only scheme given the same rapid jump; but at a different time and different radiation level.

8. line 382: with respect to my previous comment, I doubt that here is enough evidence for this.

Typos, language, etc.:

1. line 10: do you mean "initial fog droplet number concentration"?

2. line 30-31:This sentence appears trivial for ACP and can be removed.

3. throughout text and equations: put index letters in non-italic font unless they are variables themselves

4. line 154: "potentially"?!

5. Figure 1:label on y-axis for (a) should be $s_{max}$ instead of $S_{max}$ (to be in line with the nomenclature)

6. line 215: What is MONC?
* * *

---

## Referee Comment (RC2) · Anonymous Referee #2 · 27 Oct 2020

Recommendation: Major Revision

General comments: The manuscript proposes an improvement of aerosol activation parametrization for LES of fog. This is an interesting topic as most of LES of fog now use 2-moment microphysical schemes and also produce an overestimation of cloud concentration and mass, leading to a too rapid transition into optically thick fog. Whilst the topic is important, I feel major modifications to the manuscript are required, with substantial inputs essential for publication. To be frank I hesitated with a rejection.

My major concerns are that:

- The improvement brought by the SMOD scheme is not convincing, and it is a problem as it is the main objective of the paper. The authors argued a more realistic approach

but it is a theotical point of view. In fact there is no improvement for IOP1, and all the simulations show significant differences with the observations.

- The initialization of aerosol is chosen to limit the discrepancy with the observations but is not realistic, and not representative of the clean air typically found at Cardington. Indeed, direct observations of aerosol concentrations were not available for this case, but we can rely on typical measurements over Cardington. The authors would argue that the initialization of a single accumulation mode with 100 cm-3 has already been used in Poku et al. (2019) but it was already unrealistic. The reference paper for IOP1 for the time being is Boutle et al. (2018), who proposed an aerosol distribution initialised with 1000 cm-3 concentration of Aitken-mode aerosols, 100 cm-3 accumulation-mode aerosols and 2 cm-3 coarse-mode aerosols. Authors also argued that Aitken mode aerosol can be ignored, but they have to prove it by running a simulation with the 3 modes and the initial concentrations proposed by Boutle et al, and by comparing it with the present simulation. I am not at all convinced by this equivalence, and by the justification to neglect the Aitken mode. This comparison is required for the acceptance of the paper.

- In the same way, the fog life cycle is only presented during 10 hours, but it would be more interesting to present the whole life cycle until 12 UTC as in Boutle et al. (2018): we can indeed suppose that it is not shown as the simulations would depart too far from the observations.

- Too few observations are used to evaluate the simulations: other temporal evolutions, for temperature, vertical velocity variances, sensible heat flux are available for this IOP, as well as vertical profiles of cloud mixing ratio, droplet concentration, droplet diameter. New comparisons need to be added in order to prove that SMOD is more realistic. They are also required for the acceptance of the paper.

- The study is not presented in a larger context where major papers have introduced some advances in the activation parametrization. Hence Thouron et al. (2012) for

stratocumulus and Lebo et al. (2012) for deep convective clouds studied the relevance of saturation adjustment for LES. Then Schwenkel and Maronga (2019) studied the activation parametrization for LES of fog, but the authors only mention this last study in the conclusion. The necessity to consider the radiation cooling in the supersaturation evolution equation for fog is not new. So my general question is: what does this paper bring compared to the previous studies ? If new results are indeed shown, then they should be presented in the context of these other studies. It would be also necessary to compare the SMOD equations with the equations of Schwenkel and Maronga (2019) for instance.

More minor concerns are:

- For the visibility calculation, why not to use a direct calculation according to the Koschmieder (1925) equation, linking the visibility to an extinction coefficient function of the DSD, through the Mie theory, instead of a diagnostic from Gultepe et al. (2006), which could be questionable ?

- MONC needs to be presented.

- l 224-225 : the remark about the grid spacing is hardly understandable: why is it critical to run MONC at 1m or 2m resolution? The same sentence has been written in Poku et al. (2019) and it was already misunderstood.

- The radiation scheme SOCRATES is called every 5 min. Is it not too large for a LES of radiation fog, with a necessary accurate estimation of radiative cooling ?

- l 254 : it is said that radiative cooling is the biggest source of saturation. But it would be nice to compare it to the total temperature tendency, or to show that the consideration of the turbulent contribution to the non-adiabatic temperature tendency does not change the results.

- l 320 : there is a reference to the impact of surface heterogeneities, but it is not clear why this discussion is introduced here. Bergot et al. (2015) considered buildings,

while Mazoyer et al. (2018) and Ducongé et al. (2020), which considered trees and orography (over Lanfex) heterogeneities respectively, could be added.

References : - Ducongé, L., C.Lac, B.Vié, T.Bergot, and J.D. Price, Fog in heterogeneous environments : The relative importance of local and non‑local processes on radiative‑advective fog formation, Quart. J. Roy. Meteor. Soc., 146, 2522-2546, 2020. - Lebo, Z. J., Morrison, H., & Seinfeld, J. H. (2012). Are simulated aerosol-induced effects on deep convective clouds strongly dependent on saturation adjustment?. Atmospheric Chemistry and Physics, 12(20), 9941-9964. - Thouron, O., J.-L. Brenguier, and F.ăBurnet, Supersaturation calculation in large eddy simulation models for prediction of the droplet number concentration, Geosci. Model Dev., 5, 761-772, 2012.

---

## Author Comment (AC1) · 18 Mar 2021

**Is a more physical representation of aerosol activation needed for simulations of fog? - Author's response**

Responding authors: Craig Poku et al.

March 18, 2021

We would like to thank both reviewers for the detailed and constructive feedback. We have found it beneficial to receive these reviews, as it's allowed us to address weaknesses in the original manuscript and provide new focus on our key scientific outcomes.

**1 Reviewer 1's comments and responses**

**1. line 13-14: the sentence reads odd. The minimum updraft velocity threshold overpredicts the droplet number in comparison to a cooling rate? What you are saying is that a cooling rate underpredicts a droplet number. How can a cooling rate/threshold predict something?**

Thank you for this suggestion. We have reworded this line in our abstract to say the following: "Our offline model results show that using the equivalent cooling rate associated with the minimum updraft velocity threshold assumption can overpredict the droplet number by up to 70% in comparison to a typical cooling rate found in fog formation." This can be located on line 13 in our manuscript.

**2. Introduction: You elaborate the current state of fog simulation in which often a bulk microphysics scheme is used, nowadays often with aerosol activation parameterization. However, you have missed important recent work that avoids aerosol activation parameterizations by using a Lagrangian cloud model (see Schwenkel and Maronga (2020), https://www.mdpi.com/2073-4433/11/5/466). I do think it would be worth to discuss their paper in the introduction. How do your results relate to their LES? Would a direct comparison of the same case make sense? The same authors also recently published a related article in ACP (https://acp.copernicus.org/articles/19/7165/2019/acp19-7165-2019.html). You do cite this close to the end of the manuscript, but I would say their work is so closely related to what you did that I would expect that you put your work into context of the two papers in the introduction.**

Thank you for making us aware of these papers. Based on your comments, we have expanded our introduction to include more recent literature, including Schwenkel and Maronga (2020), that focuses

on avoiding aerosol activation schemes when investigating fog using LES. We have also discussed the eqs. used in Schwenkel and Maronga (2019) and demonstrated how they can relate to our work. These additions can be found from lines 73-79 and lines 95-97.

**3. Eq 6: What is the variable "L"? I think you do not define it at all.**

We have defined $L$ as the specific latent heat of vaporisation in Eq. 6 of the revised manuscript.

**4. Table 2 / case definition. To be honest, i had serious difficulties to follow your text at some places because the case definition appears a little chaotic. The is best seen in Table 2. Example: Case C_adiabatic has two tests, but for T_ship_ad has all aerosol modes and all environments. So how many simulations were performed? It is very difficult to read the table that way. It would be beneficial to list all simulations in a more straight forward order.**

Thank you for this comment. When we went through the manuscript, we could see how the original table appeared confusing. As a result, we have rewritten this table and split it into Table's 2 and 3. These tables now state that for the offline box model runs, we did in total 24 unique tests.

**5. In the legend of Figure 1 you refer to "Aitken, accumulation, and coarse model aerosols". However, if you are not an expert in aerosol physics you might not know what these modes are. And this is the first time these terms appear in the manuscript! Please provide an explanation to the reader at a suitable location. It might help to give a schematic diagram showing these modes.**

Thank you for this comment. We have added a section about aerosol modes in the introduction to include the following: "The aerosol population is split by size categories. These size categories (hereafter known as modes) are technically defined as: the Aitken mode, where the diameter, $d$, of an aerosol particle is $< 0.1$ $\mu$m; the accumulation mode, where $0.1 \leq d \leq 1.0$ $\mu$m; and the coarse mode, where $d > 1.0$ $\mu$m (Whitby, 1978). Due to their size, Aitken mode aerosols have an increased tendency to coagulate with other particles and not activate in their own right. In contrast, accumulation and coarse mode aerosols can activate into fog droplets, therefore indirectly impacting the cloud's microphysical structure and its life span (e.g. Twomey, 1974; Albrecht, 1989)." These sentences can be located from lines 35-40.

**6. line 224: How can you motivate a grid spacing without given the grid spacing? Furthermore, you cite studies that clearly state that 1 m grid spacings is best choice, but you use only 2 m. Given the very limited horizontal domain, I have to ask: why did you use such "coarse" grid spacing? It does not make sense to me.**

Thank you for pointing this out. Having revised the manuscript, we're aware that this statement (as it was mentioned in our previous study) is quite vague. Therefore, we conducted some tests that focused on changing the horizontal resolution by 1 m (high res) and 4 m (low res). When using a $w_{min}$ of 0.1 m s$^{-1}$, we found that there was no appreciable difference between the choice in horizontal resolution. However, the choice in horizontal resolution becomes important when $w_{min}$ is lowered to 0.01 m s$^{-1}$. Figure 1

[Figure]

Figure 1: Time series of: (a) - mean visibility (m) at a 2 m altitude; (b) - the liquid water path (g m$^{-2}$); (c) - the mean CDNC (cm$^{-3}$) at a 2 m altitude; (d) - the maximum updraft velocity (m s$^{-1}$) at a 2 m altitude. Purple – T_shipway_0.01; green – T_high_res_wmin_0.01; red – T_high_res_wmin_0.01; light blue – observations of (a) near-surface visibility and (b) liquid water path respectively; black dashed line - fog threshold of 1 km ; grey dashed line - $w_{min}$ threshold of 0.1 m s$^{-1}$.

shows that increasing our horizontal resolution leads to an increase in both liquid water and FDNC, leading to the fog deepening faster. The increased resolution led to more resolved vertical motions and more activation. Therefore, it would be a suitable option for us to consider a higher resolution. However, running our case at a 1 m resolution led to both an increase in computational expense. In addition, the storage required to run all the simulations required for our study would mean that we could have run into further technical complications. Finally, our work shows that the relevant difference between activation scheme is greater than horizontal resolution. Therefore, by running all of our cases at a 2 m resolution, we feel that this is the best compromise to not alter the fog evolution too greatly and in addition, not compromise the conclusions that we made for our work.

Given these changes, we have changed the wording in our paper to the following:

"Although previous studies such as Maalick et al. (2016) and Maronga and Bosveld (2017) have run LES fog simulations at higher horizontal resolutions, we found that running our cases at 2 m although for us to address our objectives, whilst compromising on both data storage and computational expense (not shown)." This is located on lines 248-251.

**7. This is one of my major concerns. Your paper is supposed to illustrate that the SMOD scheme gives a smoother/delayed transition to deep fog and thus is more realistic. But what is almost completely missing is evidence for this. First of all, you show the LANFEX observations, but you do not really compare your LES results against them. And what I see is actually, that all runs performed are way off from the observations. So how can you conclude that the SMOD scheme is more realistic? It appears a rather academic idea that is not proven by better agreement with observations for instance (you see rather bad agreement e.g in Figures 3, 5, and 7). Second, where can we see this smoother transition to deep fog? The longwave downwelling radiation is a good indicator. Looking at Figure 7, however, it looks like the fog layer in general remains less thick, no matter what scheme is used. The transitions, however, happen at the same time. And even worse: in the LES all runs go into deep fog mode quickly, while the observations indicate shallow fog or no fog until 21:00, and then a rapid thickening. I would say the Shipway scheme here is the only scheme given the same rapid jump; but at a different time and different radiation level.**

Thank you for this feedback. The main objective of the paper is to show that using an aerosol activation scheme designed for convective clouds is unsuitable for modelling radiation fog. Hence why with our MONC simulations, we focused on changing $w_{min}$ and understanding SMOD was sensitive to $r_e$. However, based on both reviewers 2 comments, we felt that addressed our original objectives too narrowly, consequently leading to our analysis not utilising the IOP1 observation dataset. With this in mind, with the evidence originally presented, SMOD did not appear to be the ideal represented for fog simulations. Given all of this reasoning, we have expanded both our analysis and discussion to address these concerns.

To begin, we firstly present a new analysis of the modelled to observed sensible heat flux (SHF), which can be located at Fig. 3b of our revised manuscript. For all our simulations, they are mostly zero or

slightly negative up until 2100 UTC, which agrees well with observations. After 2100 UTC, they all become more positive, with both T_SMOD and T_Shipway_wmin having the highest increase in SHF with time. With all of our simulations, SMOD with an increased $r_e$ of 20 $\mu$m (T_SMOD_er_20) is closest to observations, despite it still being positive after 2100 UTC. This result indicates that T_SMOD_er_20 with the default settings being amended for a change in FDNC is the best option, even though the fog still grows too much in optical thickness in comparison to observations. Next, we have added new analysis which makes use of the vertical FDNC that was observed at different time frames throughout the night (Fig. 4 in the revised manuscript). We first note that for all the tests that run with just Shipway and SMOD in their default settings, they all have the deepest fog layers. In addition, T_SMOD and T_Shipway_wmin both have the highest proportion of modelled to observed fog droplets (see Table 6). T_Shipway_0.01 initially has the best spatial vertical variation when compared to observations. However, T_SMOD_er_20 appears to perform best in comparison to observations when all three-time frames are considered throughout the night. Finally, we have amended the figures and analysis to directly compare the SMOD and Shipway tests better. We believe it is now clearer that SMOD produced better results compared with the observations from the case study.

We note that throughout this study all of our tests either 1. don't capture the initial variation shown in specific diagnostics e.g. near-surface visibility (Fig. 3a of the original manuscript) or 2. transition to too deep of a layer too fast. As discussed in Poku et al. (2019), it was suggested that given that the way to better capture the fog's transition is to just use an aerosol activation scheme that accounted for more realistic physical processes. What our study has highlighted, however, is that there are potential discrepancies that we have not accounted for in terms of aerosol-fog interaction representation. For example, Schwenkel and Maronga (2020) discussed how the use of a bulk microphysics scheme can account for the condensation rate being too high, leading to too thick of a fog layer. We also note that there are studies that have addressed to account for processes such as nucleation scavenging when simulating aerosol-cloud interactions (e.g. Miltenberger et al., 2018). Given both of these examples may be of importance to our work, we have added this section into both our analysis and conclusions for this work.

In conclusion, we have highlighted that a scheme based on adiabatic cooling is wrong, and demonstrated the magnitude of the error this can introduce both offline and online. Our new scheme is therefore more physically realistic and we proposed that it should be adopted (in models such as NWPs). However, we do not claim that this will solve all the problems with modelling fog onset and development, and indeed this is support by our more detailed comparison with the IOP observations.

**8. line 382: with respect to my previous comment, I doubt that here is enough evidence for this.**

Please refer to point 7, where we address this comment.

**9. line 10: do you mean "initial fog droplet number concentration"?**

We have amended this wording, which can be found on line 10 of our revised manuscript.

**10. line 30-31:This sentence appears trivial for ACP and can be removed.**

We have removed this sentence.

**11. throughout text and equations: put index letters in non-italic font unless they are variables themselves**

Thank you for this comment. We have gone through the manuscript and amended our index letters to be put into non-italic fonts.

**12. line 154: "potentially"?!**

We've changed this line in our revised manuscript to include the following: "and hence provide a potential solution for fog modelling that may require some form". This can be located on line 172.

**13. Figure 1:label on y-axis for (a) should be $s_{max}$ instead of $S_{max}$ (to be in line with the nomenclature)**

We have amended Fig. 1 to account for this change.

**14. line 215: What is MONC?**

We have included a description for MONC to include the following: "This section will investigate the impact that aerosol activation representation will have on fog evolution, using the Met Office Natural Environment Research Council Cloud (MONC) model (Brown et al., 2015, 2018). MONC is a large-eddy simulation model designed to research and develop parameterisations used in the forecast model. MONC and has the same equation set as the older Met Office Large Eddy Model (LEM; Gray et al., 2001) and unlike the LEM, MONC has been designed to couple with other modules, including the Cloud AeroSol Interactive Microphysics scheme (CASIM; Grosvenor et al., 2017; Miltenberger et al., 2018) and the Suite of Community Radiative Transfer codes (SOCRATES; Edwards and Slingo, 1996). MONC is widely used in the UK atmospheric science community, and has been used to study atmospheric processes in low level clouds in West Africa (Dearden et al., 2018), fog (Poku et al., 2019) and idealised convection simulations (Böing et al., 2019)." This can be located from lines 233-241.

**2  Reviewer 2's comments and responses**

**1. The improvement brought by the SMOD scheme is not convincing, and it is a problem as it is the main objective of the paper. The authors argued a more realistic approach but it is a theoretical point of view. In fact there is no improvement for IOP1, and all the simulations show significant differences with the observations.**

Thank you for your feedback. We do agree that this was the paper's main objective, however, our original manuscript was too narrow in both our analysis and discussion. Based on both yours and reviewer 1's comments, we have supported our argument by providing a more detailed comparison with IOP1 observations and expanded our discussion to account for highlighted discrepancies in our work. For a more detailed response, please refer to point 7. in Reviewer 1's comments.

[Figure]

Figure 2: (a) Maximum supersaturation, $s_{max}$ (%), against the total cooling rate for IOP1 assumed aerosol distribution with the Shipway scheme. (b) - (d) A plot of activated aerosol concentration, $N_{act}$ ($cm^{-3}$) against the total cooling rate for Aitken, accumulation and coarse mode aerosols respectively for IOP1 assumed aerosol distribution.

**2. The initialization of aerosol is chosen to limit the discrepancy with the observations but is not realistic, and not representative of the clean air typically found at Cardington. Indeed, direct observations of aerosol concentrations were not available for this case, but we can rely on typical measurements over Cardington. The authors would argue that the initialization of a single accumulation mode with 100 $cm^{-3}$ has already been used in Poku et al. (2019) but it was already unrealistic. The reference paper for IOP1 for the time being is Boutle et al. (2018), who proposed an aerosol distribution initialised with 1000 $cm^{-3}$ concentration of Aitken-mode aerosols, 100 $cm^{-3}$ accumulation-mode aerosols and 2 $cm^{-3}$ coarse-mode aerosols. Authors also argued that Aitken mode aerosol can be ignored, but they have to prove it by running a simulation with the 3 modes and the initial concentrations proposed by Boutle et al. (2018), and by comparing it with the present simulation. I am not at all convinced by this equivalence, and by the justification to neglect the Aitken mode. This comparison is required for the acceptance of the paper.**

Thank you for your feedback. However, we believe that based on the assumed aerosol-size distribution used by Boutle et al. (2018), our key conclusions would not change. Figure 2a shows the change in $s_{max}$ with cooling rate, with Figure's 2a showing the change in $N_a$ with cooling rate for Aitken, accumulation and coarse mode aerosols respectively. Within the tested parameter space, the greatest proportion of $N_a$ is within the accumulation mode. More specifically at the tested parameter's space maximum, accumulation mode $N_a$ makes up 97.2% of the total aerosol population from the assumed aerosol activation

spectrum. In our paper, our results show that the changes in activation are stronger in the offline box model in comparison to MONC. Therefore, in the context of our study, it's unlikely that running MONC in a multi-mode setting would invalidate the presented results.

We acknowledge that in our work, there are some limitations in aerosol treatment due to MONC not having a prognostic for supersaturation and MONC using a bulk microphysics scheme to calculate condensation. In both of these instances using a multi-mode aerosol spectrum may be critical, which should be the subject for future work. We have added this argument to the paper's discussion section and included the following line in the methodology:

"With regards to aerosol sizes, only accumulation mode aerosol where 0.1 $\mu$m < CCN size diameter < 1 $\mu$m) are accounted for. During IOP1, there were no direct aerosol or CCN measurements. Therefore, a CCN value of 100 cm$^{-3}$ in the accumulation mode was set, with a total soluble mass of 2.7ng throughout the initialised vertical profile and an assumed lognormal size distribution with a standard deviation of 2.0, based on typical measurements for a clean rural site similar to Cardington, UK (Boutle et al., 2018; Poku et al., 2019). Our simulations used a single aerosol mode to maintain consistency with the tests in the Shipway Box Model, which showed that when considering aerosol activation the activated $N_a$ for IOP1 can be accounted for by accumulation mode aerosols (not shown). Therefore, we believe that using a multi-mode aerosol spectrum would have led to an unnecessary computational expense in this study. This reasoning may be different should these simulations have been run with a prognostic for supersaturation, however, this is outside the scope of this work." This amendment can be found from lines 263-271.

**3. In the same way, the fog life cycle is only presented during 10 hours, but it would be more interesting to present the whole life cycle until 12 UTC as in Boutle et al. (2018): we can indeed suppose that it is not shown as the simulations would depart too far from the observations.**

Thank you for this suggestion. We agree that presenting the full life cycle to 12 UTC would have provided an interesting insight regarding different aerosol impacts on the fog evolution. At the time of this work being done, MONC was not coupled to an interactive land surface scheme. Given the discrepancies we found in our analysis of variables such as surface heat fluxes, we have focused on the earlier stages of fog develop. In addition, as we didn't have any aerosol measurements from IOP1 to compare with our study, this may have led to us coming to conclusions that do not necessarily correctly represent these impacts. Finally, despite using a more "realistic" aerosol activation scheme, our proposed set up doesn't capture features such as the fog variability correctly. This highlights the need to account for additional aerosol impacts, e.g. aerosol-radiation interactions, which would be key during the daytime phase of the fog cycle. Unfortunately, investigating these science areas are outside the scope of this work. To address this point, we have included the following in the conclusion of this paper:

"Finally, our study has focused on the first 10 hours of IOP1 and hence has not accounted for the fog evolution during daytime. Given the impact additional processes such as aerosol-radiation interactions and

interactive surface scheme will have on fog dissipation, it's critical to ensure schemes such as SOCRATES and CASIM are coupled for this future work." This addition can be located on lines 484-486.

**4. Too few observations are used to evaluate the simulations: other temporal evolutions, for temperature, vertical velocity variances, sensible heat flux are available for this IOP, as well as vertical profiles of cloud mixing ratio, droplet concentration, droplet diameter. New comparisons need to be added in order to prove that SMOD is more realistic. They are also required for the acceptance of the paper.**

Thank you for your feedback. We have compared our simulations to more available observations from IOP1 with the aim to demonstrate that unsuitability of using a scheme designed for convective clouds in fog modelling studies. We have addressed this concern in more detail in point 1 of your review and in point 7. in Reviewer 1's comments.

**5. The study is not presented in a larger context where major papers have introduced some advances in the activation parametrization. Hence Thouron et al. (2012) for stratocumulus and Lebo et al. (2012) for deep convective clouds studied the relevance of saturation adjustment for LES. Then Schwenkel and Maronga (2019) studied the activation parametrization for LES of fog, but the authors only mention this last study in the conclusion. The necessity to consider the radiation cooling in the supersaturation evolution equation for fog is not new. So my general question is: what does this paper bring compared to the previous studies ? If new results are indeed shown, then they should be presented in the context of these other studies. It would be also necessary to compare the SMOD equations with the equations of Schwenkel and Maronga (2019) for instance.**

Firstly, we would like to thank you for providing us with new literature that allowed us to put our work into the context of activation parameterisation studies. In addition to expanding both the introduction, analysis and discussion, we have clarified that our study is new in the sense that:

1. Although as suggested by Boutle et al. (2018) that the solution is to remove the $w_{min}$ threshold when simulating radiation fog, our results show that this is not necessarily a suitable option. This is highlighted with T_shipway_0.01, which although did initially perform better than the rest of the tests discussed, it transitioned to a thicker fog than both T_SMOD_er_15 and T_SMOD_er_20. As aerosol activation in T_shipway_0.01 was driven by just an updaft velocity, this suggests why a more physical based activation scheme such as SMOD is critical to simulate nocturnal radiation fog. More specifically, the choice in activation scheme is key when the fog layer may transition to an optically thick layer.

2. Although there has been studies investigating the use of the non-adiabatic framework in fog simulations, this is the first study to the author's knowledge that critiques this framework with a fog case that formed in a clean aerosol regimes (accumulation CCN $< 100$ cm$^{-3}$ as defined by Boutle et al., 2018), therefore supporting previous literature on the topic of aerosol-fog interactions.

We are aware that regarding SMOD, there are some limitations in the scheme e.g. the use of a bulk

microphysics scheme and not using a prognostic for supersaturation. However, given that our study focused on simulating a thin fog case, we feel that that it best highlights what missing physical processes are needed to be accounted for when simulating fog. In addition, whilst we've used LES in our work, the study was initially motivated to develop a new scheme suitable to be used in models such as NWPs. Therefore, we believe that we have been able to present a suitable option when looking to improve rural fog forecasts that are common in countries such as the UK. We have included these clarifications in the revised manuscript in both the Discussion and Conclusion section.

**6. For the visibility calculation, why not to use a direct calculation according to the Koschmieder (1925) equation, linking the visibility to an extinction coefficient function of the DSD, through the Mie theory, instead of a diagnostic from Gultepe et al. (2006). (2006), which could be questionable?**

Thank you for this comment. We believe that this is a good point and that it is something we are partly addressing in future with the use of the new VERA visibility diagnostic including aerosol effects, but it is beyond the scope of this work. In the revised manuscript this is only one aspect of the validation with observations, and since none of the simulations perfectly reproduce the observations we don't feel at a more accurate visibility calculation would alter the conclusions.

**7. MONC needs to be presented.**

Please refer to point 14 of Reviewer's 1 comments and responses.

**8. l 224-225 : the remark about the grid spacing is hardly understandable: why is it critical to run MONC at 1m or 2m resolution? The same sentence has been written in Poku et al. (2019) and it was already misunderstood.**

Please refer to point 6 of Reviewer's 1 comments and responses.

**9. The radiation scheme SOCRATES is called every 5 min. Is it not too large for a LES of radiation fog, with a necessary accurate estimation of radiative cooling ?**

Thank you for pointing this out. Upon further inspection, it was found that SOCRATES was being called every 30 secs as opposed to every 5 mins, which was an error on our part. However, to check this value was sufficient, a test was run with SMOD which called for SOCRATES every 10 secs. Upon comparison, there was no appreciable difference between both runs (not shown).

We've corrected this mistake in the manuscript which can be found on line 252.

**10. l 254 : it is said that radiative cooling is the biggest source of saturation. But it would be nice to compare it to the total temperature tendency, or to show that the consideration of the turbulent contribution to the non-adiabatic temperature tendency does not change the results.**

Thank you for this comment. When we were doing this study, we did consider how changes in subgrid mixing may be important for our simulations. In our work, we demonstrate that the total CDNC is

sensitive to the resolved updraft velocity's strength. The strength of the resolved updraft velocity is determined by the model's mixing scale length, $\lambda_m$, such that:

$$\lambda_m = c_s \times \max(\Delta x, \Delta y), \tag{1}$$

where $c_s$ is the Smagorinsky constant and $\max(\Delta x, \Delta y)$ is the maximum grid box size in the horizontal. Any motions smaller than $\lambda_m$ is calculated by the subgrid parameterisation, which can account for motions such as diffusion and small scale turbulent mixing.

Porson et al. (2011) discussed the importance of $c_s$ on the fog layer's development. They showed that reducing $c_s$ and hence $\lambda_m$ resulted in an increase level of TKE that was resolved, leading to the modelled boundary layer to deepen. Our work uses an aerosol activation scheme as opposed to a fixed droplet number used in the study by Porson et al. (2011), leading to the suggestion that the calculated CDNC is also sensitive to the levels of resolved TKE. Consequently, the initial fog formation may be more sensitive to the level of subgrid mixing, in addition to the change in sedimentation rate, as $w_{min}$ is reduced.

To understand the impact of subgrid mixing on the formation and development of IOP1, we had a set of tests to understand how sensitive is the fog's evolution to the choice in mixing length. We addressed this objective by changing the default $c_s$ used in test T_shipway_0.01, by doubled and halved it to 0.46 and 0.115 respectively (default Smagorinsky constant is set to $c_s = 0.23$). Until 1900 UTC, there is no appreciable change in the near-surface visibility or LWP (Figure 3a). From 1900 UTC, increasing (decreasing) $c_s$ results in an increase (decrease) in near-surface visibility. As the LWP is not appreciably different in these sensitivity tests (Figure 3b), the change in near-surface visibility is due to the amount of TKE being resolved, therefore directly impacting the maximum updraft velocity and hence CDNC (Figure 3d and c respectively). However, although the fog's evolution shows a slight sensitivity to subgrid mixing, its development appears to be mostly driven by a change in sedimentation rate due to a decrease in $w_{min}$. Given these results, we are confident that although we're not using the full non-adiabatic tendency, we are confident that our conclusions would not change in the revised manuscript. These figures have not been included in the revised manuscript.

**11. l 320 : there is a reference to the impact of surface heterogeneities, but it is not clear why this discussion is introduced here. Bergot et al. (2015) considered buildings, while Mazoyer et al. (2017) and Ducongé et al. (2020), which considered trees and orography (over Lanfex) heterogeneities respectively, could be added.**

We introduced this discussion to show how processes such as nucleation scavenging may be critical to capture the fog's spatial variability, which has been highlighted since using a more "realistic" aerosol activation scheme. We have clarified this sentence from line 375.

[Figure]

Figure 3: Time series of: (a) - mean visibility (m) at a 2 m altitude; (b) - the liquid water path (g m$^{-2}$); (c) - the mean CDNC (cm$^{-3}$) at a 2 m altitude; (d) - the maximum updraft velocity (m s$^{-1}$) at a 2 m altitude. Purple – T_shipway_0.01; green – T_shipway_wmin_0.01_double_mixing; red – T_shipway_wmin_0.01_half_mixing; light blue – observations of (a) near-surface visibility and (b) liquid water path respectively; black dashed line - fog threshold of 1 km ; grey dashed line - $w_{min}$ threshold of 0.1 m s$^{-1}$.

[revised manuscript text omitted]

---

## Author Comment (AC2) · 18 Mar 2021

The comment was uploaded in the form of a supplement:
https://acp.copernicus.org/preprints/acp-2020-904/acp-2020-904-AC2-supplement.pdf

---

## Referee Report (RR1)

**Review 2 of ACP-2020-904 : « Is a more physical representation of aerosol activation needed for simulations of fog?" by Craig Poku et al.**

Recommendation: Minor Revision

**General comments**: Based on reviewer comments, the paper has been significantly improved by providing a more detailed comparison with IOP1 observations and expanded the discussion to account for highlighted discrepancies in the work.
However a few last concerns already raised need to be better introduced before acceptation and I insist on it.

My concerns are that:

- Concerning the initialization of aerosol I still do not agree that a single accumulation mode with 100 cm$^{-3}$ is representative of typical measurements for a clean rural site similar to Cardington. In Boutle et al. (2018), it is clearly said that aerosol distribution representative of the clean air typically found at Cardington is 1000 cm$^{-3}$ concentration of Aitken-mode aerosols, 100 cm$^{-3}$ accumulation-mode aerosols and 2 cm$^{-3}$ coarse-mode aerosols. I understand that you have not the possibility for the moment to use a multi-mode aerosol spectrum which is perfectly admissible. But I am not at all convinced that considering 100 cm$^{-3}$ accumulation-mode aerosols is equivalent as you rely on tests not shown. Therefore you have to say that: i) an aerosol distribution of 1000 cm$^{-3}$ concentration of Aitken-mode aerosols, 100 cm$^{-3}$ accumulation-mode aerosols and 2 cm$^{-3}$ coarse-mode aerosols  as proposed and used in  Boutle et al. (2018) would be better representative of the clean air typically found at Cardington  but cannot be used in this paper; ii) the assumption of a single accumulation mode with 100 cm$^{-3}$ probably limits the overestimation of droplet concentration that would lead to a too rapid transition to a thick fog layer.

- You have not answered to my previous question:
For the visibility calculation, why not to use a direct calculation according to the Koschmieder (1925) equation, linking the visibility to an extinction coefficient function of the DSD, through the Mie theory, instead of a diagnostic from Gultepe et al. (2006), which could be questionable ?

- Line 470: The reference Thouron et al. (2012) for stratocumulus needs to be cited for the prognostic supersaturation in the same way as Lebo et al. (2012) for deep convective clouds.

**Reference** :

- Thouron, O., J.-L. Brenguier, and F. Burnet, Supersaturation calculation in large eddy simulation models for prediction of the droplet number concentration, *Geosci. Model Dev.*, *5*, 761-772, 2012.

---

## Author Response (AR2)

**Is a more physical representation of aerosol activation needed for simulations of fog? - Author's response**

Responding authors: Craig Poku et al.

April 8, 2021

We would to thank the reviewers again for the positive response to our revised manuscript. We've worked through the additional suggested comments to improve the paper's overall clarity.

**1 Reviewer comments and responses**

**Concerning the initialization of aerosol I still do not agree that a single accumulation mode with 100 cm$^{-3}$ is representative of typical measurements for a clean rural site similar to Cardington. In Boutle et al. (2018), it is clearly said that aerosol distribution representative of the clean air typically found at Cardington is 1000 cm$^{-3}$ concentration of Aitken-mode aerosols, 100 cm$^{-3}$ accumulation-mode aerosols and 2 cm$^{-3}$ coarse-mode aerosols. I understand that you have not the possibility for the moment to use a multi-mode aerosol spectrum which is perfectly admissible. But I am not at all convinced that considering 100 cm$^{-3}$ accumulation-mode aerosols is equivalent as you rely on tests not shown. Therefore you have to say that: i) an aerosol distribution of 1000 cm$^{-3}$ concentration of Aitken-mode aerosols, 100 cm$^{-3}$ accumulation-mode aerosols and 2 cm$^{-3}$ coarse-mode aerosols as proposed and used in Boutle et al. (2018) would be better representative of the clean air typically found at Cardington but cannot be used in this paper; ii) the assumption of a single accumulation mode with 100 cm$^{-3}$ probably limits the overestimation of droplet concentration that would lead to a too rapid transition to a thick fog layer.**

Thank you for this comment. Based on your feedback, we have rewritten this section discussing aerosol initialisation profiles to say the following:

"During IOP1, there were no direct aerosol or CCN measurements. Therefore, we initially planned to use a multi-mode lognormal aerosol distribution of 1000 cm$^{-3}$ Aitken-mode aerosols (mean diameter 0.05 $\mu$m), 100 cm$^{-3}$ accumulation-mode aerosols (mean diameter 0.15 $\mu$m) and 2 cm$^{-3}$ coarse-mode aerosols (mean diameter 1 $\mu$m), each following a standard deviation of 2.0, as proposed and used in Boutle et al. (2018). Using these values would therefore being representative of the clean air typically found at Cardington. However, our simulations used a single accumulation aerosol mode to maintain consistency

with the tests in the Shipway Box Model, which showed that when considering aerosol activation, the activated $N_a$ for IOP1 can be accounted for by accumulation mode aerosols (not shown). A consequence of assuming a single accumulation mode potentially limits droplet concentration overestimation, which would lead to the fog layer transitioning too quickly in optical thickness. However, based on our offline test results, we believe that using a multi-mode aerosol spectrum would have led to an unnecessary computational expense in this study. This reasoning may be different should these simulations have been run with a prognostic for supersaturation, but this is outside the scope of this work. To reduce computational expense and data storage, 1D diagnostics are output every 1min and 3D diagnostics are output every 5min."

This rephrased section can be found on line 262.

**For the visibility calculation, why not to use a direct calculation according to the Koschmieder (1925) equation, linking the visibility to an extinction coefficient function of the DSD, through the Mie theory, instead of a diagnostic from Gultepe et al. (2006), which could be questionable?**

Thank you for this comment. We believe that this is a good point and that it is something we are partly addressing in future with the use of the new VERA visibility diagnostic scheme. The VERA scheme is being developed at the Met Office and is an upgrade to the original visibility diagnostic formulated by Clark et al. (2008). The main benefit of using VERA is that it can account for additional aerosol processes as opposed to just changes in aerosol mass. However, as VERA is still being tested, using it in this study is beyond the scope of this work. In the revised manuscript, visibility is only one aspect of the validation with observations, and since none of the simulations perfectly reproduce the observations we don't feel that a more accurate visibility calculation would alter the conclusions.

**Line 470: The reference Thouron et al. (2012) for stratocumulus needs to be cited for the prognostic supersaturation in the same way as Lebo et al. (2012) for deep convective clouds.**

We have now referenced Thouron et al. (2012) as suggested.

**References**

Boutle, I., Price, J., Kokkola, H., Romakkaniemi, S., Kudzotsa, I., Kokkola, H., and Romakkaniemi, S. (2018). Aerosol-fog interaction and the transition to well-mixed radiation fog. *Atmos. Chem. Phys*, 18:7827–7840.

Clark, P. A., Harcourt, S. A., Macpherson, B., Mathison, C. T., Cusack, S., and Naylor, M. (2008). Prediction of visibility and aerosol within the operational Met Office Unified Model. I: Model formulation and variational assimilation. *Quarterly Journal of the Royal Meteorological Society*, 134(636):1801–1816.

Gultepe, I., Müller, M. D., Boybeyi, Z., Gultepe, I., Müller, M. D., and Boybeyi, Z. (2006). A New Visibility Parameterization for Warm-Fog Applications in Numerical Weather Prediction Models. *Journal of Applied Meteorology and Climatology*, 45(11):1469–1480.

Lebo, Z. J., Morrison, H., and Seinfeld, J. H. (2012). Are simulated aerosol-induced effects on deep convective clouds strongly dependent on saturation adjustment? *Atmospheric Chemistry and Physics*, 12(20):9941–9964.

Thouron, O., Brenguier, J. L., and Burnet, F. (2012). Supersaturation calculation in large eddy simulation models for prediction of the droplet number concentration. *Geoscientific Model Development*, 5(3):761–772.